# SurfDesign: Effective Protein Design on Molecular Surfaces

## Abstract

Structure-based inverse folding has been extensively explored in recent years. In contrast, surface-conditioned protein generation is still an under-explored area. Molecular surfaces characterized by a compact and smooth composition of atoms at their boundary hold a more direct relevance to biomolecular interactions and function. In this work, we introduce a novel framework named SurfDesign with several key improvements. Firstly, considering the theoretical fact that the molecular surface is a continuous manifold with infinite resolution, we propose surface-based equivariant message passing (SEMP) to incorporate the normal vector and curvatures and get aware of the manifold's Euclidean locality. Besides, a hybrid parameter-efficient fine-tuning (PEFT) technique is employed to combine the knowledge of protein language models (PLMs) with the surface geometric encoder. We extensively evaluate SurfDesign on the CATH, TS50, TS500, and PDB datasets, achieving an average recovery of more than 70%. Our work opens another road to designing functional proteins, underscoring the importance of including surface attributes in protein discovery.

## 1 Introduction

Proteins, as intricately folded chains of amino acids, are fundamental to biological processes such as transcription, translation, signaling, and cell cycle control. The advent of generative deep learning (DL) (Huang et al., 2016; Song et al., 2020; Rives et al., 2021) has revolutionized protein design, shifting the focus away from traditional physics-based methods (see Figure 1). One prevalent approach is to first design a target backbone structure and then identify a sequence that folds into this backbone. Despite the significant progress (Ingraham et al., 2019; Jing et al., 2020; Dauparas et al., 2022; Hsu et al., 2022; Gao et al., 2022a; Mao et al., 2023; Zheng et al., 2023; Wu & Li, 2024a; Qiu et al., 2024; Wang et al., 2024), the goal of protein design goes be-

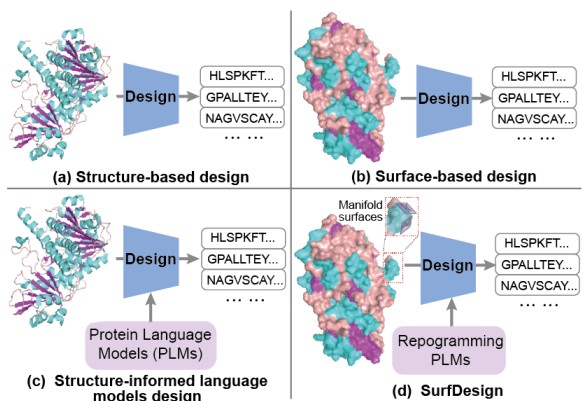

Figure 1: Prevailing setups of protein design, conditioned on backbone structures or molecular surfaces.

yond predicting a sequence that folds into a target backbone (Defresne et al., 2021). The ultimate goal is to design proteins with desired functions, such as enzymes binding to specific substrates or proteins inhibiting given targets. The inverse folding method has limitations, as it only specifies geometric constraints through the backbone structure. To achieve desired functions, it is essential to impose biochemical property constraints as well. For instance, two proteins with complementary shapes may still not bind effectively due to poorly placed charges, polarity, or hydrophobicity at their binding interface (Gainza et al., 2023).

Recent research (Song et al., 2024) has made strides in addressing this issue by designing functional proteins based on continuous surfaces augmented with biochemical properties. Albeit deep genera-

tive models show revolutionary capacity in this field, the current neural surface-conditioned protein design still has undeniable flaws in devising more plausible proteins due to the limited expression capabilities of their algorithms. First and foremost, in theory, molecular surfaces are continuous and smooth 3D manifolds with infinite resolution (Lee et al., 2023; Sun et al., 2024). Existing point cloud- (Song et al., 2024; Zhang et al., 2023) or mesh-based methods do not account for inherent connectivity and smoothness, treating surfaces as collections of discrete points. However, ideally, molecular surfaces are continuous, meaning no gaps or discrete points, and allow for differentiable operations. Besides, the smoothness of manifolds indicates a well-defined tangent space at each point and can be described using smooth functions. Secondly, the limited availability of experimentally determined protein surface data impedes progress in surface-conditioned design. For instance, the known protein structures in the commonly-used CATH (Orengo et al., 1997) dataset are vastly outnumbered by the sequence data in the UniRef (Suzek et al., 2015) sequence database. This disparity presents a challenge for data-hungry generative models, which struggle to comprehensively explore the protein sequence space and often produce sub-optimal sequence predictions. Moreover, from a biological perspective, molecular surfaces alone may not provide sufficient information, especially in buried regions where sequential knowledge is more valuable yet largely neglected.

To address these challenges, we propose SurfDesign, a novel and effective algorithm for surface-conditioned protein design (see Figure 2). SurfDesign captures the continuity and smoothness of surface manifolds by analyzing the tangent space and curvatures near each point, where normal vectors are used to approximate local geometry and curvatures are leveraged to measure deviations from planarity. We then compute directional information between neighboring points and introduce a surface-based equivariant message passing (SEMP) scheme to integrate manifold-specific geometries such as curvatures and directionality. Moreover, inspired by recent advances in employing pre-trained protein language models (PLMs) for versatile protein design (Zheng et al., 2023; Gao et al., 2023; Qiu et al., 2024; Wang et al., 2024; Mao et al., 2023), we propose a hybrid parameter-efficient fine-tuning (PEFT) technique to enhance our SEMP with the knowledge from PLMs. Comprehensive experiments have been conducted to evaluate our SurfDesign in the domain of inverse folding. Our algorithm exhibits a substantial performance boost over current state-of-the-art methods, VFN-IF (Mao et al., 2023), KW-Design (Gao et al., 2023), and InstructPLM (Qiu et al., 2024), by a large margin, achieving **74.13**% and **72.14**% sequence recovery on CATH 4.2 and 4.3 for single-chain monomers. SurfDesign has also been trained on the entire PDB database with an impressive recovery of **81**%. These results highlight SurfDesign's superior performance and potential in advancing the field of protein design. Discussion on related works is put in Appendix B.

**Problem Statement.** Neural structure-conditioned protein design aims to find the amino acid sequence $\mathcal{S} = \{s_i \in \mathrm{Cat}(20) : 1 \leq i \leq n\}$ folding into the desired structure $\mathcal{X} = \{\boldsymbol{x}_i \in \mathbb{R}^{4 \times 3} : 1 \leq i \leq n\}$, where $s_i$ belongs to one of the 20 residue types and $\mathcal{X}$ denotes the spatial coordinates for 4 backbone atoms (*i.e.*, $C_\alpha$, $C$, $N$ and $O$). It can be formulated as an end-to-end graph-to-sequence learning problem with a parameterized encoder-decoder neural network $\mathcal{F}_{\boldsymbol{\vartheta}} : \mathcal{X} \to \mathcal{S}$. Surface-conditioned protein design is analogous to the structure-conditioned definition but generates functional proteins, which fold into expected surfaces $\mathcal{Q}$ with associated biochemical properties (Song et al., 2024). Our objective therefore transfers to learn a function $\mathcal{F}_{\boldsymbol{\vartheta}}(\cdot)$:

$$\mathcal{F}_{\boldsymbol{\vartheta}} : \mathcal{Q} \to \mathcal{S} \tag{1}$$

Given sufficient surface-sequence paired data, the learning purpose is to maximize the conditional log-likelihood $p(\mathcal{S}|\mathcal{Q}; \boldsymbol{\vartheta})$. This approach allows for designing sequences that either have the highest likelihood or are generated using sampling algorithms to ensure diversity and novelty (Zheng et al., 2023). Remarkably, homologous proteins consistently share similar surfaces (Pearson & Sierk, 2005), so the surface-conditioned design is underdetermined. In other words, the valid amino acid sequence $\mathcal{S}$ may not be unique (Gao et al., 2022a).

## 2 METHOD

### 2.1 PRELIMINARY AND BACKGROUND

**Surface Generation** The surface geometry of a protein is of crucial interest for protein-protein interaction analysis. We employ PyMol (DeLano et al., 2002) to obtain the raw molecular surface, where a probe of a certain radius ($\sim 1$ Angstrom) is moved along the protein to calculate the Solvent

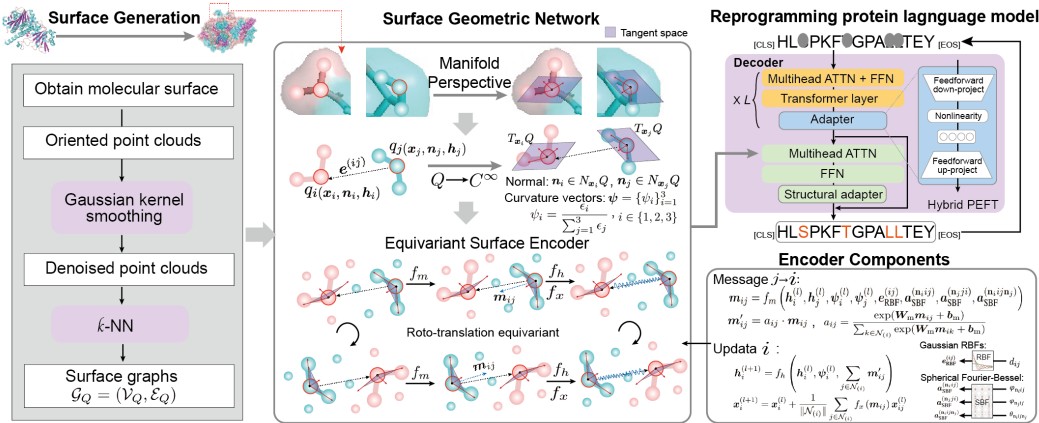

Figure 2: Illustration of SurfDesign. Smooth Surface graphs are acquired by PyMol or MSMS and processed via denoising. Then an equivariant surface encoder is appended to extract manifold representations. These features are further fed into the structural adapter of the protein language models for recovering masked amino acids.

Accessibility Surface (SAS) and Solvent Excluded Surface (SES). The consequent probe coordinates are regarded as the molecular surface, defined by an oriented point cloud $Q = \{q_i : 1 \leq i \leq m\}$ and $m >> n$. Each surface point $q_i$ has a triplet of attributes $(\boldsymbol{x}_i, \boldsymbol{n}_i, \boldsymbol{h}_i)$, where $\boldsymbol{x}_i \in \mathbb{R}^3$ and $\boldsymbol{n}_i \in \mathbb{R}^3$ is the 3D coordinates and unit normal vector, and $\boldsymbol{h}_i \in \mathbb{R}^{\phi_h}$ indicates the physicochemical properties of $q_i$ such as hydrophobicity, hbond, and charge. Then the surface graph is built via $k$-NN, resulting in $\mathcal{G}_Q = (\mathcal{V}_Q, \mathcal{E}_Q)$. Notably, we also investigate the open source MSMS (Robinson et al., 2014) and BioPython (Cock et al., 2009) for surface generation and discover ignorable differences in processing speeds among several toolkits. As raw point clouds generally carry noise and these noisy points may limit the expressivity of molecular surfaces (Alexa et al., 2001), we borrow ideas from Song et al. (2024) and apply the Gaussian kernel smoothing on raw point cloud data:

$$\boldsymbol{x}_i \leftarrow \sum_{j \in \mathcal{N}_{(i)}} \frac{\mathcal{K}(\boldsymbol{x}_i, \boldsymbol{x}_j) \cdot \boldsymbol{x}_j}{\sum_{t \in \mathcal{N}_{(i)}} \mathcal{K}(\boldsymbol{x}_i, \boldsymbol{x}_t)}, \quad \mathcal{K}(\boldsymbol{x}, \boldsymbol{y}) = \exp^{-\frac{\|\boldsymbol{x} - \boldsymbol{y}\|^2}{\eta}}, \tag{2}$$

where $\mathcal{N}_{(i)}$ denotes the neighborhood of $\boldsymbol{x}_i$ and $\mathcal{K}(.,.)$ is the Gaussian kernel with $\eta$ indicating distance scale in the point space. Here, $\eta$ is set as $\max\left(\{\|\boldsymbol{x}_i - \boldsymbol{x}_j\|^2\}_{j \in \mathcal{N}_{(i)}, i \in [m]}\right)$.

## 2.2 Surface Geometric Network

**A Manifold Perspective for Molecular Surfaces.** Theoretically, molecular surfaces are continuous manifolds with infinite resolution (Lee et al., 2023), which cannot be fully expressed by existing mesh- (Gainza et al., 2020) or point-based (Sverrisson et al., 2021; Zhang et al., 2023; Song et al., 2024) mechanisms. The key distinct property between manifold surfaces and conventional point clouds or meshes is that every point in the manifold is locally Euclidean. Mathematically, for $\forall q_i \in Q$, there exists a neighborhood $U_{q_i}$ and a homomorphism $f_{\text{homo}}(\cdot)$ such that $f_{\text{homo}} : U_{q_i} \rightarrow V \subseteq \mathbb{R}^3$, where $V$ is an open ball in $\mathbb{R}^3$. In order to describe the local geometry of a manifold point $q_i \in Q$, we need to know at least (1) the linear approximation of the manifold in its vicinity, which corresponds to *the tangent space*, and (2) how fast the surface bends or deviates from being a plane near this point, which can be measured by *curvature*.

Towards this goal, we assume that the surface $Q$ is a $C^\infty$ differentiable manifold and $T_{\boldsymbol{x}_i}Q$ denotes the tangent space of any point $\boldsymbol{x}_i \in Q$. Then we can acquire the unit normal vector $\boldsymbol{n}_i \in N_{\boldsymbol{x}_i}Q$ perpendicular to $T_{\boldsymbol{x}_i}Q$. If $Q$ is implicitly described by a signed distance function (SDF) satisfying $f_{\text{SDF}}(\cdot) = 0$, then the normal at point $\boldsymbol{x}_i$ is equivalent to the gradient, *i.e.*, $\boldsymbol{n}_i = \nabla f_{\text{SDF}}(\boldsymbol{x}_i)$. Here, we draw the normal vector set $\{\boldsymbol{n}\}_{i=1}^m$ immediately from the software (*i.e.*, PyMol) and integrate this orientation knowledge into the geometric encoder to linearly approximate the manifold and achieve manifold-awareness. Prior studies (Zhang et al., 2023; Song et al., 2024) have seldom

considered this specialty of molecular surfaces and merely handle naive clouds. One exception, dMaSIF (Strokach et al., 2020), notices this manifold uniqueness and computes the quasi-geodesic distance as $d_{ij} = \|x_{ij}\|^2 \cdot (2 - n_i^\top \cdot n_j)$ to naively resemble the geodesic coordinates in the tangent space $T_{x_i}Q$. However, its construction of tangent vectors destroys the equivariance.

Additionally, there are varying ways to define curvatures of 3D Riemannian manifolds intrinsically without reference to a larger space (Kobayashi & Nomizu, 1996), such as normal curvature $k_n$, geodesic curvature $k_g$, and geodesic torsion $\tau_r$. Those all relate the direction of curvatures to the unit normal vector $n_i$. Given a non-singular curve $\gamma(q_i) \in Q$ parametrized by arc length, we can compute $T_i = \gamma'(q_i)$ and $t_i = n_i \times T_i$ to form the Darboux frame. The triple $(T_i, t_i, n_i)$ defines a positively oriented orthonormal basis attached to each point of the curve $\gamma(q_i)$. Then the above quantities are related by $\begin{pmatrix} T' \\ t' \\ u' \end{pmatrix} = \begin{pmatrix} 0 & k_g & k_n \\ -k_g & 0 & \tau_r \\ -k_n & -\tau_r & 0 \end{pmatrix} \begin{pmatrix} T \\ t \\ u \end{pmatrix}$. Inspired by progress in geometry processing (Tian et al., 2023; Wu & Li, 2024b; Zhang et al., 2008), we estimate these quantities in a closed form from local points $\mathcal{N}_{(i)}$. Specifically, we first compute a covariance matrix for $q_i$ and its neighborhood $\mathcal{N}_{(i)}$:

$$\Sigma = \frac{1}{\|\mathcal{N}_{(i)}\|} \sum_{\mathbf{x}_j \in \mathcal{N}_{(i)}} \mathbf{x}_j \mathbf{x}_j^\top - \bar{\mathbf{x}}\bar{\mathbf{x}}^\top, \quad \Sigma \in \mathbb{R}^{3\times3}. \tag{3}$$

where $\bar{\mathbf{x}}$ is the centroid of this point cluster. Then after the eigen-decomposition of $\Sigma$ (*e.g.*, singular value decomposition or eigenvalue decomposition), eigenvalues can be attained as $\epsilon_1, \epsilon_2$, and $\epsilon_3$ ($\epsilon_1 \geq \epsilon_2 \geq \epsilon_3$). The three pseudo curvatures vectors $\psi = \{\psi_i\}_{i=1}^3$ can be therefore computed as:

$$\psi_i = \frac{\epsilon_i}{\sum_{j=1}^3 \epsilon_j}, \quad i \in \{1, 2, 3\}. \tag{4}$$

We employ $\psi$ as a substitute and approximation of the Darboux frame $(k_n, k_g, \tau_r)$. It can be proved that this curvature feature $\psi$ is roto-translation invariant (see Appendix E).

**Directionality in Surface Point Clouds.** The manifold characteristic of molecular surfaces introduces additional directional information when considering pairwise or ternary interactions among connected particles. To be specific, for each neighboring point pair $(i, j)$, two intersecting planes (see Fig. 3) is formulated with respective normals $(n_i, n_j)$. We denote the angles between normals and the connecting directed line of two points $(x_{ij}, x_{ji})$ by $\varphi_{\mathbf{n}_i ij} = \angle n_i x_{ij}$ and $\varphi_{\mathbf{n}_j ji} = \angle n_j x_{ji}$. We denote the dihedral angle between two half-phases as $\theta_{\mathbf{n}_i ij \mathbf{n}_j} = \angle n_i n_j \perp x_{ij}$. In addition to the common distance $\|x_{ij}\|^2$, these three angles provide a more comprehensive view of understanding the relative position of $(q_i, q_j)$ lying in the surface manifold $Q$, which will also be incor-

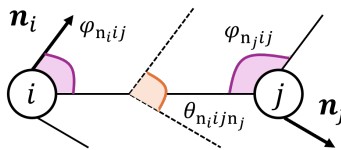

Figure 3: Angles hidden in the oriented surface point cloud, containing two intersection angles $\varphi_{\mathbf{n}_i ij} = \angle n_i x_{ij}$ and $\varphi_{\mathbf{n}_j ji} = \angle n_j x_{ji}$ as well as a dihedral angle $\theta_{\mathbf{n}_i ij \mathbf{n}_j}$.

porated into our surface modeling. For instance, for different values of $(\varphi_{\mathbf{n}_i ij}, \varphi_{\mathbf{n}_j ji}, \theta_{\mathbf{n}_i ij \mathbf{n}_j})$, a triplet of $(\frac{\pi}{2}, \frac{\pi}{2}, 0)$ indicates a perfectly smooth region, while a triplet of $(\pi, \pi, \pi)$ implies a severely sharp and steep curve.

**Equivariant Surface Encoder.** Finally, we draw inspiration from prevalent and modern equivariant algorithms (Satorras et al., 2021; Gasteiger et al., 2021; 2020b;a; Zhang et al., 2023; Song et al., 2024) and propose a **s**urface-based **e**quivariant **m**essage **p**assing (SEMP) as the encoder of $\mathcal{F}_\vartheta(\cdot)$. Our SEMP architecture is roto-translation equivariant, leveraging both directional and curvature information. To begin with, by setting an interaction cutoff $c_{\text{int}}$, we calculate the 3D spherical Fourier-Bessel bases $\left(a_{\text{SBF}}^{(\mathbf{n}_i ij)}, a_{\text{SBF}}^{(\mathbf{n}_j ji)}\right) \in 2 \times \mathbb{R}^{N_{\text{CBF}} \times N_{\text{SBF}} \times N_{\text{RBF}}}$ for two angles $\varphi \in \left[\varphi_{\mathbf{n}_i ij}^{(l)}, \varphi_{\mathbf{n}_j ji}^{(l)}\right]$ to integrate orientation knowledge between each interactive particles in the surface:

$$a_{\text{SBF},ovt}^{(l)}\left(\left\|x_{ij}^{(l)}\right\|^2, \varphi, \theta_{\mathbf{n}_i ij \mathbf{n}_j}^{(l)}\right) = \sqrt{\frac{2}{c_{\text{int}}^3 j_{o+1}^2(z_{ov})}} j_o\left(\frac{z_{ov}}{c_{\text{int}}} \left\|x_{ij}^{(l)}\right\|^2\right) Y_o^t\left(\varphi_{\mathbf{n}_i ij}^{(l)}, \theta_{\mathbf{n}_i ij \mathbf{n}_j}^{(l)}\right), \tag{5}$$

where $o \in [N_{\text{CBF}}]$, $v \in [N_{\text{SBF}}]$, and $t \in [N_{\text{RBF}}]$ control the degree, root, and order of the radial basis functions, respectively. Besides, $j_o(\cdot)$ is the $o$-th degree spherical Bessel functions and $z_{ov}$ is its corresponding $v$-th root. $Y_o^t(\cdot)$ is the real $o$-th degree and $t$-th order spherical harmonics. Equ. 5 can be boiled down to a joint 2D basis if the order $t$ is set to 0. By using $Y_o^0(\cdot)$, we obtain the 2D representation $\boldsymbol{a}_{\text{SBF}}^{(\mathbf{n}_i ij\mathbf{n}_j)} \in \mathbb{R}^{N_{\text{CBF}} \times N_{\text{SBF}}}$ based on $\theta_{\mathbf{n}_i ij\mathbf{n}_j}^{(l)}$.

Remarkably, those 2D/3D spherical Fourier-Bessel representations $\boldsymbol{a}_{\text{SBF}}^{(\mathbf{n}_i ij)}$, $\boldsymbol{a}_{\text{SBF}}^{(\mathbf{n}_j ji)}$, and $\boldsymbol{a}_{\text{SBF}}^{(\mathbf{n}_i ij\mathbf{n}_j)}$ enjoy the roto-translation invariant property due to their exploitation of the relative distance as well as the invariant angles. Then those directional vectors along with pointwise curvatures are fed into SEMP to attain the initial messages $\boldsymbol{m}_{ij}$ as:

$$\boldsymbol{m}_{ij} = f_m \left( \boldsymbol{h}_i^{(l)}, \boldsymbol{h}_j^{(l)}, \boldsymbol{\psi}_i^{(l)}, \boldsymbol{\psi}_j^{(l)}, \boldsymbol{e}_{\text{RBF}}^{(ij)}, \boldsymbol{a}_{\text{SBF}}^{(\mathbf{n}_i ij)}, \boldsymbol{a}_{\text{SBF}}^{(\mathbf{n}_j ji)}, \boldsymbol{a}_{\text{SBF}}^{(\mathbf{n}_i ij\mathbf{n}_j)} \right), \tag{6}$$

where $f_m$ is a multi-layer perception (MLP) appended with an activation function like SiLU (Nwankpa et al., 2018). $\boldsymbol{e}_{\text{RBF}}^{(ij)}$ is the radial basis function representation of the interatomic distance $\|\boldsymbol{x}_{ij}\|^2$. Then a softmax is employed to reweight the messages:

$$\boldsymbol{m}_{ij}' = a_{ij} \cdot \boldsymbol{m}_{ij}, \quad a_{ij} = \frac{\exp(\boldsymbol{W}_{\text{m}} \boldsymbol{m}_{ij} + \boldsymbol{b}_{\text{m}})}{\sum_{k \in \mathcal{N}_{(i)}} \exp(\boldsymbol{W}_{\text{m}} \boldsymbol{m}_{ik} + \boldsymbol{b}_{\text{m}})} \tag{7}$$

where the weight matrix $\boldsymbol{W}_{\text{m}} \in \mathbb{R}^{\phi_m \times 1}$ and vector $\boldsymbol{b}_{\text{m}} \in \mathbb{R}$ are learnable. After that, messages are propagated from the vicinity of each point $q_i$ to update its node feature as well as coordinates:

$$\boldsymbol{h}_i^{(l+1)} = f_h \left( \boldsymbol{h}_i^{(l)}, \boldsymbol{\psi}_i^{(l)}, \sum_{j \in \mathcal{N}_{(i)}} \boldsymbol{m}_{ij}' \right), \quad \boldsymbol{x}_i^{(l+1)} = \boldsymbol{x}_i^{(l)} + \frac{1}{\|\mathcal{N}_{(i)}\|} \sum_{j \in \mathcal{N}_{(i)}} f_x(\boldsymbol{m}_{ij}) \boldsymbol{x}_{ij}^{(l)}. \tag{8}$$

where $f_h(\cdot)$ is another MLP and $f_x : \mathbb{R}^{\phi_m} \to \mathbb{R}$ transforms $\boldsymbol{m}_{ij}$ into a scalar score to control the impact of directional vector $\boldsymbol{x}_{ij}^{(l)}$. Notably, as the position of each point $\boldsymbol{x}_i^{(l)}$ is moving as the layer $l \in [L]$ goes deeper with $\boldsymbol{x}_i^{(0)} = \boldsymbol{x}_i$, it is optional but recommended to adjust and re-calculate the curvature $\boldsymbol{\psi}_i$ and relevant angles $(\varphi_{\mathbf{n}_i ij}, \varphi_{\mathbf{n}_j ji}, \theta_{\mathbf{n}_i ij\mathbf{n}_j})$ simultaneously. As angles $(\varphi_{\mathbf{n}_i ij}, \varphi_{\mathbf{n}_j ji}, \theta_{\mathbf{n}_i ij\mathbf{n}_j})$ depend on each normal vector pair ($\boldsymbol{n}_i$ and $\boldsymbol{n}_j$), we adopt the local least fitting method (Mitra & Nguyen, 2003) to estimate and renew $\{\boldsymbol{n}_i\}_{i=1}^m$. In specific, for $q_i$'s updated coordinates $\boldsymbol{x}_i^{(l)}$ at the $l$-th layer, we compute the covariance $\boldsymbol{\Sigma}^{(l)}$ according to Equ. 3 and decompose it to obtain three sorted eigenvalues as well as their corresponding eigenvectors ($\boldsymbol{\nu}_1, \boldsymbol{\nu}_2, \boldsymbol{\nu}_3$). Then $\boldsymbol{\nu}_3$ with the least eigenvalue is selected as the normal vector $\boldsymbol{n}^{(l)}$ at the $l$-th layer.

## 2.3 REPROGRAMMING PROTEIN LANGUAGE MODELS

**PEFT for SurfDesign.** Recent works have explored the possibility of transforming PLMs (Rives et al., 2021; Lin et al., 2022; Hu et al., 2022) into protein design models, and massive evidence demonstrates that the emergent evolutionary knowledge hidden in those PLMs can vastly facilitate the structure-conditioned protein design. Concretely, LM-Design (Zheng et al., 2023), InstructPLM (Qiu et al., 2024), KW-Design (Gao et al., 2023), and VFN-IF-ESM (Mao et al., 2023) report improvements in CATH 4.2 of 10.8% (recovery $50.22\% \to 55.65\%$), 73.9% (perplexity $10.28 \to 2.68$), 14.4% (recovery $54.74\% \to 62.67\%$), and 17.6% (recovery $51.66\% \to 60.77\%$), respectively. Motivated by this progress, we also leverage PLMs as the decoder of $\mathcal{F}_\vartheta(\cdot)$ and stack several parameter-efficient fine-tuning (PEFT) techniques to fully release the potential of PLMs and significantly reduce the memory budget. Specifically, we utilize a hybrid PEFT method combined with a structural adapter (Zheng et al., 2023) and LoRA (Hu et al., 2021) with a rank of $r = 4$ and a scaling constant of $\alpha = 8$. It is worth mentioning that there is still no consensus on which sort of PEFT strategies are most suitable for PLMs (Sledzieski et al., 2024), and we practically find our hybrid mechanism more effective than a singular one for surface-conditioned protein design.

**Training.** Following LM-Design (Zheng et al., 2023), we employ the conditional masked language modeling (CMLM) to better accommodate PLMs that are tasked with MLM (Devlin et al., 2018) as the training objective. Given the surface $\mathcal{Q}$, CMLM decomposes the sequence into masked and

Table 1: Sequence design performance and ablation studies on CATH 4.2 held-out test split. The **best performance** is shown in bold, while the best baseline is indicated with an underline. ESM-IF is tested on CATH 4.2, although it was originally trained and evaluated on CATH 4.3.

| Models | Trainable/Total Params. | Perplexity (↓) | | | Median Recovery (↑) | | |
|---|---|---|---|---|---|---|---|
| | | Short | Single-chain | All | Short | Single-chain | All |
| StructGNN (Ingraham et al., 2019) | 1.4M / 1.4M | 8.29 | 8.74 | 6.40 | 29.44 | 28.26 | 35.91 |
| GraghTrans (Ingraham et al., 2019) | 1.56M / 1.56M | 8.39 | 8.83 | 6.63 | 28.14 | 28.46 | 35.82 |
| GCA (Tan et al., 2023) | 2.1M / 2.1M | 7.09 | 7.49 | 6.05 | 32.62 | 31.10 | 37.64 |
| GVP (Jing et al., 2020) | 1.0M / 1.0M | 7.23 | 7.84 | 5.36 | 30.60 | 28.95 | 39.47 |
| AlphaDesign (Gao et al., 2022b) | 3.6M / 3.6M | 7.32 | 7.63 | 6.30 | 34.16 | 32.66 | 41.31 |
| ProteinMPNN (Dauparas et al., 2022) | 1.9M / 1.9M | 6.21 | 6.68 | 4.61 | 36.35 | 34.43 | 45.96 |
| ESM-IF (Hsu et al., 2022) | 142M / 142M | 6.93 | 6.65 | 3.96 | 35.28 | 33.78 | 48.95 |
| PiFold (Gao et al., 2022a) | 6.6M / 6.6M | 6.04 | 6.31 | 4.55 | 39.84 | 38.53 | 51.66 |
| LM-Design-MPNN (Zheng et al., 2023) | 5.0M / 659M | 7.01 | 6.58 | 4.41 | 35.19 | 40.00 | 54.41 |
| LM-Design-PiFold (Zheng et al., 2023) | 11.9M / 664M | 6.77 | 6.46 | 4.52 | 37.88 | 42.47 | 55.65 |
| DPLM (Wang et al., 2024) | 5.0M / 659M | – | – | – | – | – | 54.54 |
| InstructPLM (Qiu et al., 2024) | 89.1M / 6.6B | 3.22 | 3.17 | 2.68 | 61.59 | 59.29 | 57.51 |
| KW-Design (Gao et al., 2023) | 6.4M / 798M | 5.48 | 5.16 | 3.46 | 44.66 | 45.45 | 60.77 |
| VFN-IF (Mao et al., 2023) | 5.4M / 5.4M | 5.70 | 5.86 | 4.17 | 41.34 | 40.98 | 54.74 |
| VFN-IF-ESM (Mao et al., 2023) | 5.4M / 15B | 4.92 | 4.22 | 3.36 | 50.00 | 52.13 | 62.67 |
| SurfPro (Song et al., 2024) | 5.8M / 5.8M | – | – | 3.13 | – | – | 57.78 |
| SurfDesign (w/o PLMs) | 5.3M / 5.3M | 3.21 | 3.10 | 3.08 | 62.70 | 64.88 | 65.35 |
| SurfDesign (w/o SEMP) | 4.8M / 655M | 3.08 | 2.93 | 2.76 | 65.43 | 67.06 | 66.27 |
| SurfDesign | 5.3M / 656M | **2.43** | **2.44** | **2.41** | **73.74** | **75.17** | **74.13** |

observed ones as $\mathcal{S} = \mathcal{S}_{\text{masked}} \cup \mathcal{S}_{\text{obs}}$ and assumes a conditional independence over identities of target residues $s_i \in \mathcal{S}_{\text{masked}}$. Then it requires the model to predict a set of target amino acids $\mathcal{S}_{\text{masked}}$ from the remaining observed residues $\mathcal{S}_{\text{obs}}$:

$$p(\mathcal{S}_{\text{masked}}|\mathcal{S}_{\text{obs}}, \mathcal{Q}; \theta) = \Pi_{s_i \in \mathcal{S}_{\text{masked}}} p(s_i|\mathcal{S}_{\text{obs}}, \mathcal{Q}; \theta) \qquad (9)$$

where $\mathcal{S}_{\text{masked}}$ is randomly masked. Moreover, Zheng et al. (2023) presents a coarse-to-fine manner to reconstruct a protein native sequence from its corrupted version. We also explore this inference scheme with iterative refinement (Savinov et al., 2021) but discover no benefit.

## 3 EXPERIMENTS

We evaluate SurfDesign on various benchmarks for fixed backbone protein sequence design, including single-chain monomers (Sec. 3.1) and multi-chain protein complexes (Sec. 3.2). More experimental details, dataset statistics, and additional results are elaborated in the Appendix A.

**Baselines and Datasets.** A wide variety of baseline approaches are established for a fair comparison and most of them are open source. Among them, StructGNN (Ingraham et al., 2019), GraphTrans (Ingraham et al., 2019), GVP (Jing et al., 2020), ProteinMPNN (Dauparas et al., 2022), AlphaDesign (Gao et al., 2022b), PiFold (Gao et al., 2022a), UniIF (Gao et al., 2024), and etc. are GNN-based algorithms. In contrast, DenseCPD (Qi & Zhang, 2020) is a CNN-based approach. Besides, DPLM (Wang et al., 2024), InstructPLM (Qiu et al., 2024), LM-Design (Zheng et al., 2023), KW-Design (Gao et al., 2023) and VFN-IF-ESM (Mao et al., 2023) leverage and integrate the knowledge of pretrained PLMs. SurfPro (Song et al., 2024) is a surface-based framework. Using the same splitting strategy as the compared systems (Jing et al., 2020; Dauparas et al., 2022; Gao et al., 2022a), proteins in CATH 4.2 were partitioned into 18,024/608/1,120 samples for training, validation, and testing, respectively. To compare with ESM-IF (Hsu et al., 2022), structures in CATH 4.3 were split into 16,153/1,457/1,797 samples for training, validation, and testing, separately. To provide a head-to-head comparison with ESM-IF, no extra data such as AF2DB (Varadi et al., 2022) is utilized for training SurfDesign. To evaluate the generative quality thoroughly, we report perplexity, and median recovery rate on short-chain, single-chain, and all-chain settings as usual. The multi-chan protein design employs the dataset curated by Dauparas et al. (2022), which was preprocessed by clustering sequences at 30% identity, resulting in 25,361 clusters. Following ProteinMPNN's setup, those clusters were divided randomly into 23,358/1,464/1,539 samples for training, validation, and testing, respectively. This strategy ensures that none of the chains from the target chain or biounits of the target chain were present in the other two sets.

Table 2: Sequence design on CATH 4.3. †: SINGLE-CHAIN in Hsu et al. (2022) is defined differently.

| Models | Perplexity (↓) | | | Recovery Rate (↑) | | |
|---|---|---|---|---|---|---|
| | Short | Single-chain | All | Short | Single-chain | All |
| GVP (Hsu et al., 2022) | 7.68 | †6.12 | 6.17 | 32.60 | 39.40 | 39.20 |
| ProteinMPNN (Dauparas et al., 2022) | 6.31 | 6.32 | 4.85 | 40.30 | 39.02 | 48.25 |
| ESM-IF (Hsu et al., 2022) | 8.18 | †6.33 | 6.44 | 31.30 | 38.50 | 38.30 |
|   + 1.2M AF2 Data | 6.05 | †4.00 | 4.01 | 38.10 | 51.50 | 51.60 |
| PiFold (Gao et al., 2022a) | 5.88 | 5.55 | 4.47 | 42.86 | 43.69 | 50.68 |
| VFN-IF (Mao et al., 2023) | – | – | – | 45.34 | 53.70 | 52.18 |
| UniIF (Gao et al., 2024) | – | – | – | 45.41 | 54.46 | 53.05 |
| LM-Design-MPNN (Zheng et al., 2023) | 5.88 | 5.66 | 4.19 | 45.71 | 46.15 | 56.38 |
| LM-Design-PiFold (Zheng et al., 2023) | 5.66 | 5.52 | 4.01 | 46.84 | 48.63 | 56.63 |
| KW-Design (Gao et al., 2023) | 5.47 | 5.23 | 3.49 | 43.86 | 45.95 | 60.38 |
| SurfDesign | **5.08** | **4.97** | **3.12** | **66.74** | **71.30** | **72.14** |

## 3.1 SINGLE-CHAIN PROTEIN DESIGN

**Results.** Table 1 and 2 document the results of SurfDesign in comparison to the comprehensive strong baselines on the CATH (Orengo et al., 1997) benchmark. It can be concluded that SurfDesign consistently achieves state-of-the-art performance in distinct settings. In particular, we observe that SurfDock is the foremost to exceed 70% recovery on not only CATH 4.2 but also CATH 4.3, illustrating its superior capacity in restoring effective protein sequences. Besides, on the full CATH 4.2 benchmark, SurfDesign achieves a perplexity of 2.41 and a recovery of 74.13%, outpassing the previous state-of-the-art VFN-IF-ESM (Mao et al., 2023) by 28.27% and 18.28%, separately. It also induces recovery improvements of 19.72% and 26.78% on the short and single-chain subsets, respectively. Furthermore, SurfDesign surpasses SurfPro, another surface-based algorithm, by 23.00% and 28.29% in the overall metrics, respectively. The outstanding phenomenon exists for the CATH 4.3 benchmark as well, where SurfDesign outperforms the strongest competitor KW-Design (Gao et al., 2023) by 10.60% and 19.49% for perplexity and recovery, respectively. To summarize, SurfDesign enhances surface-conditioned sequence generation with greater efficiency, thanks to the significant advancements and open-source contributions from the entire community, building on the foundation laid by previous pioneers.

## 3.2 MULTI-CHAIN PROTEIN COMPLEX DESIGN

**Results.** A protein only functions when it docks, combines, and interacts with other macro-molecules, forming multi-chain protein complexes. Therefore, studying protein sequence design for multi-chain assembled structures is crucial for drug design. This motivates us to assess whether SurfDesign can more effectively manage protein complex design. From Table 3, we conclude that the recovery is generally higher for longer proteins and all models achieve higher recovery rates on PDB than CATH datasets. More importantly, SurfDesign attains the best performance with a recovery of more than 80%. This phenomenon indicates that SurfDesign can design both single-chain proteins and multi-chain complexes. This makes SurfDesign more versatile regarding the categories and scenarios where it can be deployed, creating opportunities to use it for designing specific protein complexes.

## 3.3 ZERO-SHOT GENERALIZATION TO NEW PROTEIN FAMILIES

**Results.** TS50 and TS500 are commonly used independent test sets to assess model generalization for unseen proteins introduced by Li et al. (2014). Towards this goal, we evaluate SurfDesign trained on CATH 4.2 and 4.3 respectively and report the results in Table 4. It can be discovered that SurfDesign outpasses prior studies by a large margin on all benchmarks. Specifically, it achieves a perplexity of 2.05 and a recovery rate of 82.16 on TS50, which outperforms the previous state-of-the-art algorithm, VFN-IF-ESM, by 18.65% and 12.08%, respectively. Meanwhile, on the TS500 dataset, SurfDesign obtains a perplexity of 1.98 and a recovery rate of 84.70. These numbers are better than VFN-IF-ESM by 22.04% and 16.80%, separately. In addition, for those trained in CATH 4.3, SurfDesign consistently achieves the best. In a nutshell, SurfDesign is the pioneer to transcend 82% and 84% recovery on the TS50 and TS500.

Table 3: Performance on multi-chain protein complex dataset (*i.e.*, PDB).

| Models | Recovery (↑) | | | |
|---|---|---|---|---|
| length | $L < 100$ | $100 \leq L < 500$ | $500 \leq L < 1000$ | Full |
| StructGNN (Ingraham et al., 2019) | 0.41 | 0.41 | 0.42 | 0.41 |
| GraphTrans (Ingraham et al., 2019) | 0.40 | 0.39 | 0.40 | 0.40 |
| GCA (Tan et al., 2023) | 0.41 | 0.41 | 0.42 | 0.41 |
| GVP (Jing et al., 2020) | 0.44 | 0.42 | 0.45 | 0.43 |
| AlphaDesign (Gao et al., 2022b) | 0.48 | 0.49 | 0.50 | 0.49 |
| ProteinMPNN (Dauparas et al., 2022) | 0.52 | 0.53 | 0.55 | 0.53 |
| PiFold (Gao et al., 2022a) | 0.54 | 0.58 | 0.60 | 0.58 |
| LM-Design-MPNN (Zheng et al., 2023) | – | – | – | 0.61 |
| LM-Design-GVP (Zheng et al., 2023) | – | – | – | 0.62 |
| KWDesign (Gao et al., 2023) | 0.59 | 0.66 | 0.67 | 0.66 |
| SurfDesign | **0.74** | **0.79** | **0.82** | **0.81** |

Table 4: Performance comparison on TS50 and TS500. Following prior literature, we mainly report the results using models trained on CATH 4.2. Numbers in the brackets are results from models trained on CATH 4.3.

| Models | TS50 | | | TS500 | | |
|---|---|---|---|---|---|---|
| | Perplexity (↓) | Recovery (↑) | Worst (↑) | Perplexity (↓) | Recovery (↑) | Worst (↑) |
| DenseCPD (Qi & Zhang, 2020) | – | 50.71 | – | – | 55.53 | – |
| StructGNN (Ingraham et al., 2019) | 5.40 | 43.89 | 26.92 | 4.98 | 45.69 | 0.05 |
| GraphTrans (Ingraham et al., 2019) | 5.60 | 42.20 | 29.22 | 5.16 | 44.66 | 0.03 |
| GVP (Jing et al., 2020) | 4.71 | 44.14 | 33.73 | 4.20 | 49.14 | 0.09 |
| GCA (Tan et al., 2023) | 5.09 | 47.02 | 28.87 | 4.72 | 47.74 | 0.03 |
| AlphaDesign (Gao et al., 2022b) | 5.25 | 48.36 | 32.31 | 4.93 | 49.23 | 0.03 |
| KW-Design (Gao et al., 2023) | 3.10 | 62.79 | 39.31 | 2.86 | 69.19 | 0.02 |
| VFN-IF (Mao et al., 2023) | 3.58 | 59.54 | – | 3.19 | 63.65 | – |
| VFN-IF-ESM (Mao et al., 2023) | 2.52 | 73.30 | – | 2.54 | 72.49 | – |
| InstructPLM (Qiu et al., 2024) | 2.29 | 67.99 | – | 2.42 | 64.22 | – |
| ProteinMPNN (Dauparas et al., 2022) | 3.93 (3.62) | 54.43 (54.22) | 37.24 (41.18) | 3.53 (3.27) | 58.08 (57.23) | 0.03 (0.04) |
| PiFold (Gao et al., 2022a) | 3.86 (3.70) | 58.72 (59.68) | 37.93 (38.14) | 3.44 (3.70) | 60.42 (59.95) | 0.03 (0.05) |
| LM-Design-MPNN (Zheng et al., 2023) | 3.82 (3.60) | 56.92 (58.13) | 35.17 (39.14) | 2.13 (2.15) | 64.30 (63.76) | 0.04 (0.04) |
| LM-Design-PiFold (Zheng et al., 2023) | 3.50 (3.27) | 57.89 (61.38) | 39.74 (46.75) | 3.19 (3.09) | 67.78 (66.56) | 0.02 (0.04) |
| SurfDesign | **2.05 (2.03)** | **82.16 (83.44)** | **41.30 (47.81)** | **1.98 (1.96)** | **84.70 (85.12)** | **0.10 (0.08)** |

## 3.4 MORE RESULTS AND ANALYSIS

**Ablation Studies.** We conduct systematic experiments to investigate the contributions of different components in SurfDesign, shown in Table 1. It can be observed that the knowledge of PLMs provides a large improvement of 13.43% in recovery (65.35% → 74.13%) and a decrease of 24.29% in perplexity (3.21 → 2.43). Moreover, the incorporation of directionality and curvatures also contributes to the superiority of SurfDesign with an improvement of 11.86% in recovery and 12.68% in perplexity.

**Structural Contexts.** To further understand the action mechanism of SurfDesign, we dissect its performance according to different structural contexts in Figure 4. Structure-based LM-Design shows high recovery on structurally constrained residues in the folding core, while low recovery in structurally less constrained residues on surface areas and loops. SurfDesign significantly enhances the recovery on structurally constrained and less-constrained residues, particularly those on the surface regions.

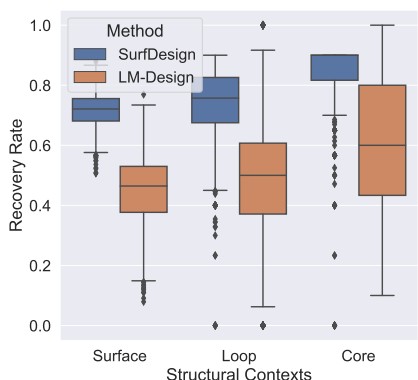

Figure 4: Comparison of sequence recovery w.r.t. structural contexts regarding SASA and interaction interface, on CATH 4.2 single-chain proteins.

**Surface Recovery.** Unlike the conventional structure-conditioned protein design, the ultimate goal of our surface-based design is to generate proteins with higher surface similarity of key regions such

Table 5: Evaluation on the surface recovery on CATH 4.2.

| Models | IoU ($\uparrow$) | CD ($\downarrow$) | NC($\uparrow$) |
|---|---|---|---|
| LM-Design | 0.90 | 5.972 | 0.4236 |
| VFN-IF-ESM | 0.92 | 4.688 | 0.4859 |
| SurfDesign | 0.98 | 2.873 | 0.6241 |

Table 6: Structure recovery comparison based on the self-consistent protocol from Yim et al. (2023). ‡: benchmarked results are quoted from Mao et al. (2023).

| Metrics | PiFold‡ | LM-Design‡ | VFN-IF-ESM‡ | SurfDesign |
|---|---|---|---|---|
| scTM $> 0.5$ | 90.98% | 89.42% | 93.29 % | 96.17% |
| scRMSD $< 2.0$ | 60.35 % | 58.41% | 64.16% | 72.83% |

as the binding or interaction site (Lai et al., 2024). In order to evaluate the similarity between two 3D molecular shapes, we follow ideas from (Sun et al., 2024) and use three evaluation metrics commonly used in 3D modeling from three aspects: volume, distance, and normal vectors. They are Volumetric Intersection over Union (IoU), Chamfer distance (CD), and Normal Consistency (NC) (computational details are in Appendix A.2). As shown in Table 5, SurfDesign can reconstruct the molecular surfaces well, which accords with the motivation of our surface-conditioned design. Visualization of generated and ground truth surfaces are available in Appendix A.3.

**Structure Recovery.** We compare SurfDesign with strong baselines in terms of protein structure recovery on CATH 4.2, reported in Table 6. Following standard evaluation procedures (Yim et al., 2023; Mao et al., 2023), ESMFold was used to predict structures of designed sequences. A case study of visualization comparison using Alphafold-3 is displayed in Appendix C. Two self-consistent metrics, scTM ($\uparrow$) and scRMSD ($\downarrow$) are leveraged to assess the similarity between desired and designed protein structures. It can be found that SurfDesign is more likely to generate protein sequences with expected structures.

**Scalability of PLMs.** The scaling law w.r.t model sizes of PLMs has recently been studied (Zheng et al., 2023; Qiu et al., 2024). To understand the influence of PLM model sizes over SurfDesign's capacity, we increase the parameters of ESM-2 from 8M to 3B. As indicated in Figure 5, a similar phenomenon has been discovered where the performance of SurfDesign improves as PLMs scale. When integrating knowledge from the largest PLM (3B), SurfDesign achieves a 76.01% recovery rate on CATH 4.2. This coincidence highlights the great potential of empowering surface-conditioned design with cutting-edge PLMs (Kaplan et al., 2020).

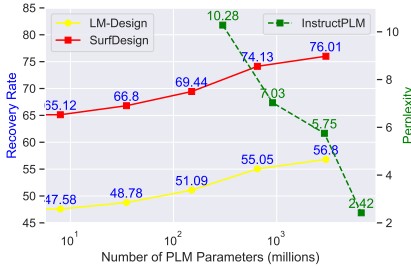

Figure 5: Performance in terms of model scales of PLMs using ESM2.

## 4 CONCLUSION

We propose SurfDesign, a novel method that integrates the geometric and biochemical information from molecular surfaces to design proteins with the knowledge of protein language models. SurfDesign is the foremost model that achieves 70% recovery on CATH 4.2, CATH 4.3, TS50, TS500, and PDB, demonstrating its generalizability and effectiveness. We look forward to future efforts in extending its application to real-world problems such as antibody and enzyme discovery.

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

## A EXPERIMENTAL DETAILS

**Training and metrics.** The models were trained up to 50 epochs by default using the Adam optimizer on 4 A100 GPUs. We used the same training settings as ProteinMPNN (Dauparas et al., 2022) and LM-Design (Zheng et al., 2023), where the batch size was set to approximate 6000 residues, and the Adam optimizer was aligned with a NOAM learning rate scheduler. Following previous works, perplexity and *median* recovery scores are reported. In Table 1 and 2, two subsets of the entire test set are also reported. Particularly, the SHORT set contains proteins up to length 100, and the SINGLE CHAIN set contains proteins recorded as a single chain in PDB (Berman et al., 2002).

**Implementation.** PyMol is adopted for surface generation in our implementation. We have tried the fast sampling algorithm introduced by dMaSIF (Sverrisson et al., 2021) and used by later studies (Wu & Li, 2024b), which approximates the protein surface as the level set of a smooth distance function. However, this sampling mechanism has unacceptable randomness and is abandoned for SurfDesign. As for the biochemical feature computation, we follow MaSIF (Gainza et al., 2020) and calculate three key invariant point inputs, including the Poisson Boltzmann electrostatics using

APBS [1], the hydrophobicity [2], and the free electrons/protons [3]. After a further ablation study, we discover that the hydrophobicity and the charge are pivot to the performance improvement while the electrostatics is not necessary.

For Figure 4, we employ RSA to determine the surface and core. To be specific, residues with RSA greater than 0.25 are considered on the surface, while residues with RSA less than 0.1 are regarded as core residues. We use the DSSP algorithm to decide the loop regions.

## A.1 DATASET INFORMATION

Table 7 documents the vertex count statistics for the CATH datasets. We observe an equal distribution over vertex in different splits. Besides, comparing our surface with SurfPro (Song et al., 2024), it can be found that our surface is more sparse with nearly half of the average vertex per residue. This difference is due to the different computation techniques adopted by various software for surface generation (*e.g.*, PyMol and MSMS).

Table 7: Vertex counts statistics for surfaces from the CATH 4.2 and CATH 4.3 datasets.

| Vertex Count | CATH 4.2 | | | CATH 4.3 | | |
|---|---|---|---|---|---|---|
| | Train | Validation | Test | Train | Validation | Test |
| Average Vertex Count Per Residue | 53.47 | 53.56 | 53.31 | 53.36 | 55.27 | 53.11 |
| Maximum Vertex Count | 27,817 | 25,614 | 25,433 | 27,110 | 27,817 | 25,968 |
| Minimum Vertex Count | 1,923 | 2,315 | 2,022 | 1,923 | 2,011 | 2,000 |
| Preprocess Time Per Protein | | 0.38s | | | 0.36s | |

## A.2 SURFACE COMPARISON

### A.2.1 EVALUATION METRICS

Motivated by DSR (Sun et al., 2024), we employ IoU, CD, and NC to assess the similarity between the molecular surfaces of designed proteins and target proteins. For simplicity, these three metrics are all normalized to a range of $0 - 1$. They provide a comprehensive evaluation of the model's performance from different perspectives and are defined as follows.

**IoU.** IoU compares the reconstructed volume with the ground truth shape (higher is better). For two arbitrary shapes $A, B \subseteq \mathbb{S} \in \mathbb{R}^n$ is attained by $\text{IoU} = \frac{|A \cap B|}{|A \cup B|}$.

**CD.** CD is a standard metric to evaluate the distance between two point sets $\mathcal{X}_1, \mathcal{X}_2 \subset \mathbb{R}^n$ (lower is better) as $d_C(\mathcal{X}_1, \mathcal{X}_2) = \frac{1}{2}\left(d_{\overrightarrow{C}}(\mathcal{X}_1, \mathcal{X}_2) + d_C(\mathcal{X}_2, \mathcal{X}_1)\right)$, where $d_{\overrightarrow{C}}(\mathcal{X}_1, \mathcal{X}_2) = \frac{1}{|\mathcal{X}_1|} \sum_{\boldsymbol{x}_1 \in \mathcal{X}_1} \min_{\boldsymbol{x}_2 \in \mathcal{X}_2} \|\boldsymbol{x}_1 - \boldsymbol{x}_2\|$.

**NC.** NC evaluates estimated surface normals (higher is better). Normal consistency between two normalized unit vectors $n_i$ and $n_j$ is defined as the dot product between the two vectors. For evaluating the surface normals, given the object surface points and normal vectors: $X_{\text{pred}} = \{(\boldsymbol{x}_i, \overrightarrow{n_i})\}$, and the ground truth surface points and normal vectors: $X_{gt} = \{(\boldsymbol{y}_j, \overrightarrow{m_j})\}$, the surface normal consistency between $X_{\text{pred}}$ and $X_{gt}$, denoted as $\Gamma$, is defined as: $\Gamma(X_{gt}, X_{\text{pred}}) = \frac{1}{|X_{gt}|} \sum_{j \in |X_{gt}|} \left| \overrightarrow{n_j} \cdot \overrightarrow{m}_{\theta(\boldsymbol{y}_j, X_{\text{pred}})} \right|$, where $\theta(\boldsymbol{y}_j, X_{\text{pred}} := \{(\boldsymbol{x}_i, \overrightarrow{n_i})\}) = \arg\min_{i \in |X_{\text{pred}}|} \|\boldsymbol{y}_j - \boldsymbol{x}_i\|_2^2$.

---

[1]`https://github.com/LPDI-EPFL/masif/blob/master/source/triangulation/computeAPBS.py`

[2]`https://github.com/LPDI-EPFL/masif/blob/master/source/triangulation/computeHydrophobicity.py`

[3]`https://github.com/LPDI-EPFL/masif/blob/master/source/triangulation/compuSurfDesignteCharges.py`

### A.3 SURFACE VISUALIZATION

In addition to protein structure restoration, we show the surface similarity between designed and ground truth proteins. We envision the surface of designed proteins and target proteins in Figure 6. A heavy overlap can be found between the point clouds of the designed protein surface and the ground truth protein surface with a pretty low CD and significantly high IoU. These all indicate that SurfDesign produces proteins with expected surface shapes.

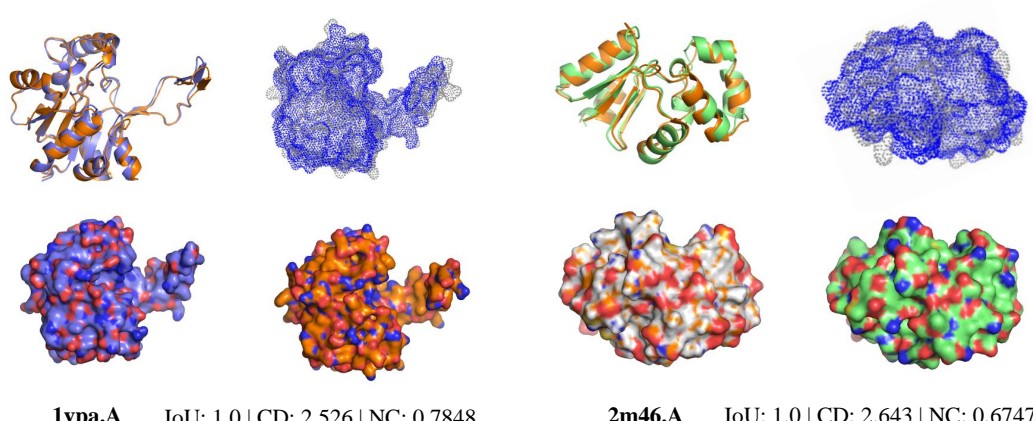

**1vpa.A**    IoU: 1.0 | CD: 2.526 | NC: 0.7848        **2m46.A**    IoU: 1.0 | CD: 2.643 | NC: 0.6747

Figure 6: Comparision between original and designed surfaces, where molecular surfaces are visualized from two perspectives: the point cloud view and the manifold view.

## B    RELATED WORK

**Structure-based Protein Design.**    The protein folding problem, a longstanding challenge in biology, has seen significant advancements through the application of AI techniques like AlphaFold (Jumper et al., 2021) and RoseTTAFold (Baek et al., 2021). The complementary problem, known as *inverse folding*, has also garnered increasing attention. The most primitive group relies on multi-layer perceptrons (MLPs) to predict the probability of 20 amino acids for each residue based on handcrafted features. SPIN (Li & Koehl, 2014) achieved a 30% recovery rate on the TS50 dataset by incorporating torsion angles, sequence profiles, and energy profiles. This was further improved by SPIN2 (O'Connell et al., 2018), which added features such as backbone angles, local contact number, and neighborhood distance, reaching a 34% recovery rate. Concurrently, other methods, like the one proposed by Wang et al. (2018), employed features including backbone dihedrals, solvent-accessible surface area (SASA) of backbone atoms, secondary structure types, and unit direction vectors, achieving a 33% recovery rate. Another category uses 2D or 3D CNN to extract protein features. For instance, SPROF (Chen et al., 2019) adopts 2D CNN to learn residue representations from the distance matrix and achieves a 40.25% recovery on TS500. ProDCoNN (Zhang et al., 2022a) designs a nine-layer 3D CNN with multi-scale convolution kernels and achieves 42.2% recovery on TS500. DenseCPD (Qi & Zhang, 2020) further enhanced recovery to 55.53% using the DenseNet architecture (Huang et al., 2017). As proteins can be natured represented as graphs, GNNs are extensively employed to consider structural constraints, with nodes and edges representing residue information and pairwise interactions, respectively. Notable works include GraphTrans (Ingraham et al., 2019), which introduced a graph attention encoder and autoregressive decoder, and GVP (Jing et al., 2020), which incorporated geometric vector perceptrons for learning from scalar and vector features. Subsequent developments enclose GCA (Tan et al., 2023), which introduces global graph attention for learning contextual features, AlphaDesign (Gao et al., 2022b), which presents a simplified graph encoder and a constraint-aware decoder based on GVP, ProteinMPNN (Dauparas et al., 2022), which capitalizes on the benefits of an auto-regressive encoding-decoding scheme and message-passing updating techniques, and PiFold (Gao et al., 2022a), which improves the traditional encoding-decoding framework by introducing virtual atoms and backbone dihedrals. Lately, VFN (Mao et al., 2023)

proffers learnable vector computations between coordinates of frame-anchored virtual atoms and exhibits an impressive 62.67% recovery.

Despite the enormous advancements, the diversity of generated sequences is limited by the small scale of training data. ESM-IF (Hsu et al., 2022) addressed this by leveraging the accurate protein folding predictions of AlphaFold2 to train a large-scale inverse folding framework using GVP. LM-Design (Zheng et al., 2023) tackles the data limitation by fine-tuning ESM models and employing embeddings from pre-trained structural encoders to recover design sequences through conditional mask prediction. Subsequently, InstructPLM (Qiu et al., 2024) utilizes the cross-modality alignment in LLMs and introduces structure prompts to fine-tune ProGen2 (Nijkamp et al., 2023). KW-Design (Gao et al., 2023) proposes a knowledge-aware module that refines low-quality residues with knowledge from ESM and GearNet (Zhang et al., 2022b). Another interesting line (Wang et al., 2024) demonstrates their self-supervised discrete diffusion probabilistic framework is versatile protein learners for tasks like structure-conditioned sequence generation.

**Protein Surface Modeling.** The characteristics of the molecular surface dictate the type and strength of the interactions that a protein can have with other molecules. It is defined by van der Waals (vdW) radii(Connolly, 1983) and is commonly represented as meshes derived from signed distance functions. MaSIF (Gainza et al., 2020) pioneered the use of mesh-based geometric DL to abstract the internal parts of the protein fold and explore protein interactions. A subsequent study (Sverrisson et al., 2021) reduced pre-computation costs by modeling molecular surfaces as point clouds with atom categories assigned to each point. Other seminal works have linked protein surfaces with structural information in a multimodal manner (Somnath et al., 2021) incorporating comprehensive pretraining strategies (Wu & Li, 2024b) using implicit neural representations (INRs) (Park et al., 2019) for self-supervised learning Lee et al. (2023) and dynamic structure modeling Sun et al. (2024). Despite these efforts, protein design based on surface features remains underexplored. Recent advancements, such as the work by Gainza et al. (2023) on expanding MaSIF for *de novo* binder design, and SurfPro (Song et al., 2024), which eliminates the need for handcrafted feature calculations, have started to address this gap by generating functional proteins directly from surface data.

**Parameter-efficient Fine-tuning for Language Models.** The development of protein language models (PLMs) has been accelerated by the availability of vast datasets of amino acid sequences (Rives et al., 2021; Lin et al., 2022; Rao et al., 2019; Elnaggar et al., 2020; Madani et al., 2020; Nijkamp et al., 2023). However, training and storing full copies of large PLMs for various downstream tasks are increasingly impractical, necessitating parameter-efficient fine-tuning (PEFT) methods. Recent works (Sledzieski et al., 2024; Zeng et al., 2023) have shown that PEFT techniques, such as LoRA (Hu et al., 2021) and prompt tuning (Lester et al., 2021), achieve competitive or superior performance compared to full fine-tuning, with significantly reduced memory requirements for tasks like protein-protein interaction prediction, signal peptide prediction, and homo-oligomer symmetry prediction. In addition, biologists attempt to incorporate structural information into PLMs using advanced PEFT tools. For example, LM-Design (Zheng et al., 2023) introduces a lightweight adapter to realize structural awareness, referred to as *structural surgery* on PLMs. SES-Adapter (Tan et al., 2024) integrates structural data by converting it into sequential vectors through tools like Fold-Seek (Van Kempen et al., 2024) and DSSP (Kabsch & Sander, 1983), enabling cross-modal attention calculations. It defeats structure-aware PLMs such as SaProt (Su et al., 2023) across standard datasets, including those for thermostability, metal ion binding, gene ontology (GO) annotations, and subcellular localization predictions.

## C    VISUALIZATION RESULTS

In this section, we visualize several protein structure restoration results of SurfDesign, as shown in Figrue 7 and Figure 8. The designed structures were obtained using the latest AlphaFold 3 (Abramson et al., 2024) [4]

---

[4]We employed the Alphafold Server for inference at `https://alphafoldserver.com/`.

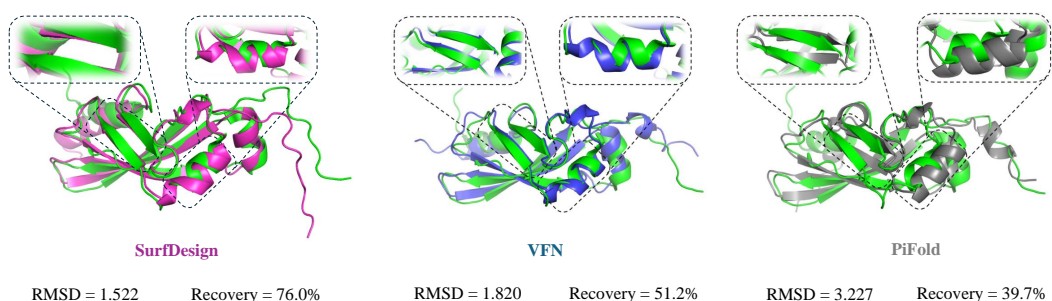

Figure 7: Visualization results of a challenging sample (PDB 2KRT). We use AlphaFold3 to recover the structure based on the predicted sequence and compare it against the experimentally determined ground-truth structure.

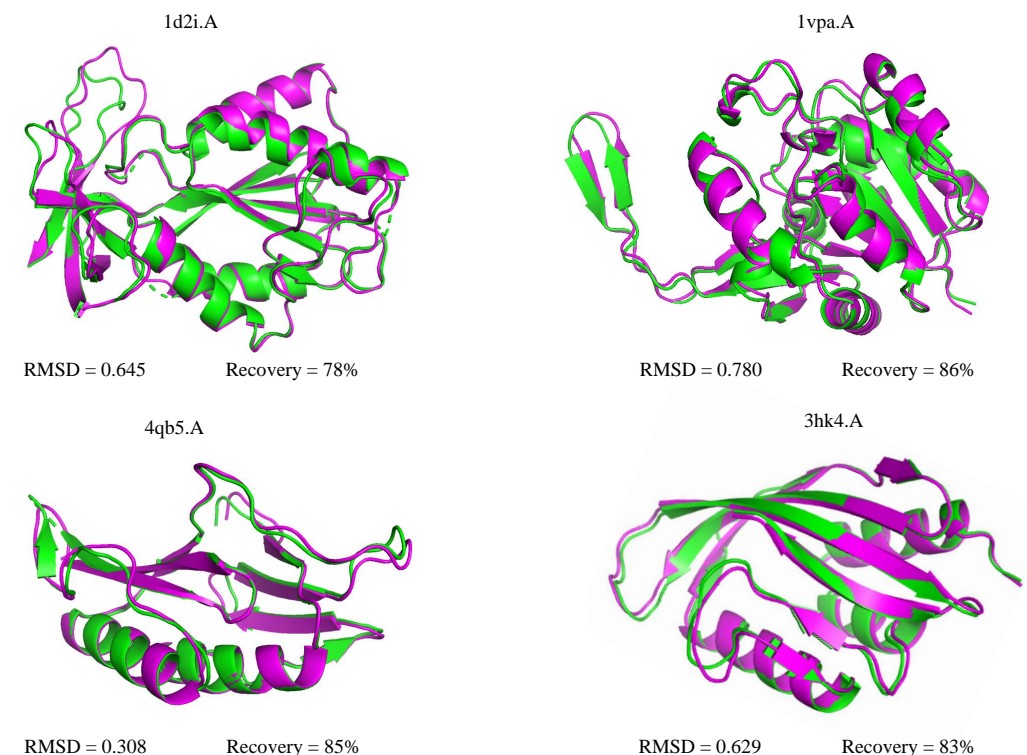

Figure 8: Visalization of SurfDesign, where the green and pink ones are ground truth and designed structures, respectively.

## D  REFOLDABILITY ANALYSIS

Following Wang et al. (2023), firstly, to assess whether the generated sequences can respect the structure condition, we evaluate the agreement of the ground truth structure with the predicted structures using the TM-score (Zhang & Skolnick, 2004). We refer to this metric as Ref-TM. Furthermore, to evaluate the folding stability of the generated sequences, we compute the mean value of the per-residue confidence estimate pLDDT predicted by the structure prediction models, referred to as Ref-pLDDT. As pLDDT is a reliable predictor of disorder (Tunyasuvunakool et al., 2021), AlphaFold2 (Jumper et al., 2021), OmegaFold (Wu et al., 2022), and ESMFold (Lin et al., 2022) are leveraged as a structure prediction model, which helps minimize deviations due to the choice of model.

We examine SurfDesign on the same 82 test samples of the CATH dataset and results are reported in Table 8. We observe that SurfDesign stands out as the leading design method across the refoldability metrics, competitive with ProteinMPNN. It achieves 0.89 Ref-TM and 89.42 Ref-pLDDT with AlphaFold2 prediction. ProteinMPNN is slightly behind with a 0.87 Ref-TM and 87.89 Ref-pLDDT, followed by LM-Design.

| Design method | ESMFold | | OmegaFold | | AlphaFold2 | | Recovery% |
|---|---|---|---|---|---|---|---|
| | TM | pLDDT | TM | pLDDT | TM | pLDDT | |
| Wildtype | 0.80 | 74.91 | 0.75 | 78.39 | 0.90 | 89.87 | 100 |
| Uniform | 0.05 | 27.68 | 0.05 | 31.53 | 0.06 | 33.68 | 5.00 |
| Natural frequencies | 0.07 | 30.53 | 0.07 | 35.59 | 0.06 | 35.02 | 5.84 |
| AF-Design | 0.53 | 61.37 | 0.53 | 72.04 | 0.52 | 75.29 | 15.95 |
| ESM-Design | 0.38 | 59.65 | 0.38 | 62.66 | 0.37 | 60.02 | 17.33 |
| StructTrans | 0.72 | 68.85 | 0.64 | 70.35 | 0.79 | 80.66 | 35.89 |
| GVP | 0.73 | 69.67 | 0.67 | 74.33 | 0.83 | 84.29 | 39.46 |
| ProteinMPNN | 0.80 | 76.53 | 0.76 | **80.75** | 0.87 | 87.89 | 41.44 |
| PiFold | 0.71 | 67.55 | 0.64 | 70.21 | 0.82 | 82.54 | 44.86 |
| LM-Design | 0.73 | 72.12 | 0.70 | 77.58 | 0.85 | 87.26 | 51.23 |
| SurfDesign | **0.81** | **79.35** | **0.76** | 80.11 | **0.89** | **89.42** | **70.19** |

Table 8: Refoldability metric and recovery metric on the CATH dataset. We employ **bold** and underline to highlight the best and suboptimal results on each metric. We use TM and pLDDT to represent Ref-TM and Ref-pLDDT.

In addition to the amino acid recovery rate, we have incorporated Foldable Diversity and sc-TM as recommended in  to further verify the diversity and self-consistency of the generated sequences. Foldable Diversity evaluates only those sequence pairs that are structurally consistent with the input protein backbone, providing a more targeted diversity metric that avoids penalizing high-quality, diverse designs. Self-consistency TM score (sc-TM), following [D], gauges the consistency of structural predictions for generated sequences, leveraging a fixed threshold of $TM_{\min} = 0.7$ as implemented by [B]. We refer to https://github.com/flagshippioneering/pi-rldif for computation, and the results are shown below. The analysis shows that SurfDesign maintains high structural consistency with competitive diversity, outperforming other methods on foldable diversity metrics and providing substantial evidence of the model's capability to generate high-quality, diverse sequences that remain faithful to the structural constraints of input proteins.

| Dataset Model | Foldable Diversity ↑ | sc-TM ↑ |
|---|---|---|
| ProteinMPNN (T=0, RD) | 20% | 0.80 |
| ProteinMPNN (T=0.1) | 23% | 0.67 |
| ProteinMPNN (T=0.2) | 3% | 0.30 |
| ProteinMPNN (T=0.3) | 0.1% | 0.14 |
| PiFold (T=0.1) | 23% | 0.72 |
| PiFold (T=0.2) | 8% | 0.38 |
| KWDesign (T=0.1) | 18% | 0.79 |
| KWDesign (T=0.2) | 23% | 0.58 |
| SurfDesign | **23**% | **0.84** |

Table 9: Foldable diversity on CATH-all.

# E  MATHEMATICAL ANALYSIS

Here we demonstrate that the curvature feature $\psi$ is roto-translation invariant. Firstly, suppose we translate the entire neighborhood $\mathcal{N}_{(i)}$ by a vector $\mathbf{t} \in \mathbb{R}^3$, so each point $\mathbf{x}_j \in \mathcal{N}_{(i)}$ is transformed

to $\mathbf{x}'_j = \mathbf{x}_j + \mathbf{t}$. - When computing the covariance matrix $\boldsymbol{\Sigma}$, the centroid $\overline{\mathbf{x}}$ is subtracted from each point in $\mathcal{N}_{(i)}$. The centroid after translation becomes $\overline{\mathbf{x}}' = \overline{\mathbf{x}} + \mathbf{t}$, so the translated covariance matrix becomes:

$$\boldsymbol{\Sigma}' = \frac{1}{\left\|\mathcal{N}_{(i)}\right\|} \sum_{\mathbf{x}_j \in \mathcal{N}_{(i)}} \left(\mathbf{x}_j + \mathbf{t}\right)\left(\mathbf{x}_j + \mathbf{t}\right)^\top - \overline{\mathbf{x}}'\overline{\mathbf{x}}^\top. \tag{10}$$

Expanding this, we get:

$$\boldsymbol{\Sigma}' = \frac{1}{\left\|\mathcal{N}_{(i)}\right\|} \sum_{\mathbf{x}_j \in \mathcal{N}_{(i)}} \mathbf{x}_j\mathbf{x}_j^\top + \mathbf{t}\mathbf{t}^\top + 2\mathbf{t} \cdot \sum_{\mathbf{x}_j \in \mathcal{N}_{(i)}} \mathbf{x}_j^\top / \left\|\mathcal{N}_{(i)}\right\| - (\overline{\mathbf{x}} + \mathbf{t})(\overline{\mathbf{x}} + \mathbf{t})^\top. \tag{11}$$

This simplifies back to the original $\boldsymbol{\Sigma}$ since $\mathbf{t}$ terms cancel out in the computation of $\boldsymbol{\Sigma}$ after translating by $\mathbf{t}$. Therefore, the covariance matrix $\boldsymbol{\Sigma}$ is invariant under translations.

Suppose we apply a rotation $\mathbf{R} \in \mathrm{SO}(3)$ to all points in $\mathcal{N}_{(i)}$, where $\mathbf{R}$ is an orthogonal matrix with determinant 1. Then each point $\mathbf{x}_j \in \mathcal{N}_{(i)}$ is transformed to $\mathbf{x}'_j = \mathbf{R}\mathbf{x}_j$. The centroid $\overline{\mathbf{x}}$ also transforms under the rotation, so the new centroid is $\overline{\mathbf{x}}' = \mathbf{R}\overline{\mathbf{x}}$. The covariance matrix $\boldsymbol{\Sigma}'$ after rotation becomes:

$$\boldsymbol{\Sigma}' = \frac{1}{\left\|\mathcal{N}_{(i)}\right\|} \sum_{\mathbf{x}_j \in \mathcal{N}_{(i)}} \mathbf{R}\mathbf{x}_j \left(\mathbf{R}\mathbf{x}_j\right)^\top - \overline{\mathbf{x}}'\overline{\mathbf{x}}'^\top. \tag{12}$$

Expanding the terms, we obtain:

$$\boldsymbol{\Sigma}' = \mathbf{R}\left(\frac{1}{\left\|\mathcal{N}_{(i)}\right\|} \sum_{\mathbf{x}_j \in \mathcal{N}_{(i)}} \mathbf{x}_j\mathbf{x}_j^\top - \overline{\mathbf{x}}\overline{\mathbf{x}}^\top\right)\mathbf{R}^\top = \mathbf{R}\boldsymbol{\Sigma}\mathbf{R}^\top. \tag{13}$$

Since a rotation is a similarity transformation, the eigenvalues of $\boldsymbol{\Sigma}'$ are the same as those of $\boldsymbol{\Sigma}$. Therefore, the eigenvalues $\epsilon_1, \epsilon_2$, and $\epsilon_3$, which are used to compute $\psi$, remain unchanged under rotations.

