# OpenReview forum: "SurfDesign: Effective Protein Design on Molecular Surfaces"
_ICLR.cc/2025/Conference — Submitted to ICLR 2025_

### Official Review · Reviewer_Gipq · 2024-11-01

**Soundness:** 3
**Presentation:** 3
**Contribution:** 2
**Rating:** 6
**Confidence:** 5

**Summary:**

This paper proposes a function-focusing inverse-folding approach utilizing surface features, advancing beyond the traditional structure-focusing inverse-folding methods that primarily emphasize structure consistency. The authors introduce an equivariant message passing technique designed to effectively process surface features, demonstrating improved sequence recovery compared to existing models.

**Strengths:**

The paper presents an innovative equivariant message passing approach capable of effectively representing surface features.

**Weaknesses:**

### Insufficient Metrics for Functionality Preservation

1. The improved sequence recovery observed in this model compared to existing approaches like PiFold or ProteinMPNN is an expected outcome, given the inclusion of surface features closely related to side-chains that determine residues, unlike previous models that only consider backbone atoms.
2. Relying solely on global accuracy metrics such as recovery (accuracy) is insufficient to claim the model's novelty.
3. Given that surface functionality, the focus of this paper, is primarily local information, it is crucial to evaluate recovery specifically for local functional sites.
4. The metrics presented in Table 5 are geometrical and do not adequately represent physically and chemically meaningful insights.
5. Additional experiments are necessary to demonstrate that newly designed sequences maintain similar functionality:
   - For instance, when designing surface-specific sequences for hydrophobic sites, the new sequences should maintain hydrophobic properties while exhibiting sequence diversity.
   - Failure to achieve this would indicate that the model has not fulfilled its motivation of function-specific design.
   - Case studies or novel metrics demonstrating successful design for specific functional sites are needed like [1] or [2].

[1] Lee, Youhan, and Jaehoon Kim. "ShapeProt: Top-down Protein Design with 3D Protein Shape Generative Model." bioRxiv (2023): 2023-12.
[2] Ektefaie, Yasha, et al. "Reinforcement learning on structure-conditioned categorical diffusion for protein inverse folding." arXiv preprint arXiv:2410.17173 (2024).

### Limited Analysis of Multi-chain Complex Dataset
1. For complex design, a more detailed analysis specific to interaction sites is required.
2. A comprehensive examination of how well the new sequences preserve the functionality of the initial complex surface while proposing diverse sequences is necessary.

### Lack of Computational Efficiency Comparison
1. The paper does not provide a comparison of overall processing times between methods.
2. Surface feature calculation may introduce significant computational overhead not present in other models. A time comparison would better illustrate the trade-offs between the proposed approach and existing methods.

**Questions:**

### Insufficient Metrics for Functional Preservation
1. Can you provide case study or suggest a new metric on local functionality design using your model? and compare with other models?

### Limited Analysis of Multi-chain Complex Dataset
1. Can you analyze how your surface-based inverse-folding model outperforms other models on the interface design of protein complexs?

### Lack of Computational Efficiency Comparison
1. Can you provide the comparison on the speed of whole sequence design protocol including data processing and inference?

---

> ### Author Response · Authors · 2024-11-13
> **A Kind Rebuttal to Reviewer Gipq**
>
> Thank you for your thoughtful review and detailed feedback. We appreciate your valuable insights, which highlight areas for further improvement and help us enhance our model's utility and rigor. We address each of your points below.
>
> (1) **Insufficient Metrics for Functional Preservation:** Thank you for your insightful suggestion on introducing metrics to evaluate SurfDesign’s effectiveness in designing sequences for specific functional regions. Following your guidance, we have integrated metrics from [A] and [B] to provide a more comprehensive assessment of SurfDesign’s performance.
>
> **Structural Compatibility and Stability (Ref-TM and Ref-pLDDT):** Following [A] and PDB-Struct [C], we have incorporated two structure-aware metrics to evaluate how well SurfDesign preserves structural integrity and stability. Ref-TM measures how closely the generated sequences align with the ground truth structure through the TM-score, assessing SurfDesign’s ability to respect structural conditions. Ref-pLDDT provides a per-residue folding stability score for generated sequences, with values based on AF2’s pLDDT confidence. On the same CATH test samples used in [C], SurfDesign achieves outstanding results: a Ref-TM of 0.89 and a Ref-pLDDT of 89.42, demonstrating structural robustness on par with ProteinMPNN (which achieves a Ref-TM of 0.87 and Ref-pLDDT of 87.89). These results substantiate SurfDesign's effective, generalized design capabilities beyond mere memorization, supporting robust performance across diverse structures.
>
> | Design method | ESMFold |  | OmegaFold |  | AlphaFold2 |  | Recovery% |
> | :---: | :---: | :---: | :---: | :---: | :---: | :---: | :---: |
> |  | TM | pLDDT | TM | pLDDT | TM | pLDDT |  |
> | Wildtype | 0.80 | 74.91 | 0.75 | 78.39 | 0.90 | 89.87 | 100 |
> | Uniform | 0.05 | 27.68 | 0.05 | 31.53 | 0.06 | 33.68 | 5.00 |
> | Natural frequencies | 0.07 | 30.53 | 0.07 | 35.59 | 0.06 | 35.02 | 5.84 |
> | AF-Design | 0.53 | 61.37 | 0.53 | 72.04 | 0.52 | 75.29 | 15.95 |
> | ESM-Design | 0.38 | 59.65 | 0.38 | 62.66 | 0.37 | 60.02 | 17.33 |
> | StructTrans | 0.72 | 68.85 | 0.64 | 70.35 | 0.79 | 80.66 | 35.89 |
> | GVP | 0.73 | 69.67 | 0.67 | 74.33 | 0.83 | 84.29 | 39.46 |
> | ProteinMPNN | $\underline{0.80}$ | $\underline{76.53}$ | $\underline{0.76}$ | $\mathbf{80.75}$ | $\underline{0.87}$ | $\underline{87.89}$ | 41.44 |
> | PiFold | 0.71 | 67.55 | 0.64 | 70.21 | 0.82 | 82.54 | 44.86 |
> | LM-Design | 0.73 | 72.12 | 0.70 | 77.58 | 0.85 | 87.26 | $\underline{51.23}$ |
> | SurfDesign | $\mathbf{0.81}$ | $\mathbf{79.35}$ | $\mathbf{0.76}$ | $\underline{80.11}$ | $\mathbf{0.89}$ | $\mathbf{89.42}$ | $\mathbf{70.19}$ |
>
> **Foldable Diversity and Self-consistency (Foldable Diversity and sc-TM):** In addition to the amino acid recovery rate, we have incorporated Foldable Diversity and Self-consistency TM score (sc-TM) as recommended in [B] to further verify the diversity and self-consistency of the generated sequences. Foldable Diversity evaluates only those sequence pairs that are structurally consistent with the input protein backbone, providing a more targeted diversity metric that avoids penalizing high-quality, diverse designs. Self-consistency TM score (sc-TM), following [D], gauges the consistency of structural predictions for generated sequences, leveraging a fixed threshold of $TM_{min}=0.7$ as implemented by [B]. We refer to https://github.com/flagshippioneering/pi-rldif for computation, and the results are shown below (numbers of all baselines are copied from [B] and we thank authors for their contributions). The analysis shows that SurfDesign maintains high structural consistency with competitive diversity, outperforming other methods on foldable diversity metrics and providing substantial evidence of the model’s capability to generate high-quality, diverse sequences that remain faithful to the structural constraints of input proteins.
>
> | Dataset | Model | Foldable Diversity $\uparrow$ | sc-TM $\uparrow$ |
> | :--- | :--- | :---: | :---: |
> |  | ProteinMPNN (T=0, RD) | 20% | 0.80 |
> |  | ProteinMPNN (T=0.1) | 23% | 0.67 |
> |  | ProteinMPNN (T=0.2) | 3% | 0.30 |
> |  | ProteinMPNN (T=0.3) | 0.1% | 0.14 |
> | CATH-all | PiFold (T=0.1) | 23%| 0.72 |
> |  | PiFold (T=0.2) | 8% | 0.38 |
> |  | KWDesign (T=0.1) | 18% | 0.79 |
> |  | KWDesign (T=0.2) | 23% | 0.58 |
> | | SurfDesign (Ours) |  **23%** | **0.84** |
>
> [A] ShapeProt: Top-down Protein Design with 3D Protein Shape Generative Model. arXiv 2023.12.
>
> [B] Reinforcement learning on structure-conditioned categorical diffusion for protein inverse folding. arXiv 2024.10.
>
> [C] PDB-Struct: A Comprehensive Benchmark for Structure-based Protein Design. arXiv 2023
>
> [D] Diffusion probabilistic modeling of protein backbones in 3d for the motif-scaffolding problem. ICLR 2022.
>
> Thank you for prompting these enhancements, which have allowed us to highlight SurfDesign’s advantages in both structural stability and diversity, underscoring its suitability for functional and structurally informed sequence design.

---

> > ### Author Response · Authors · 2024-11-13
> > **A Kind Rebuttal to Reviewer Gipq (Continued)**
> >
> > (2) **Limited Analysis of Multi-chain Complex Dataset:**
> > Thank you for noting this aspect. To provide a more in-depth evaluation of multi-chain complexes, we conducted additional experiments to assess SurfDesign’s performance on complex interface design tasks. Specifically, we evaluated the generated sequences for their ability to preserve interaction-site functionality while introducing sequence diversity. Three additional metrics are included: **Binding site ratio (BSR)** compares the overlap between the binding sites of the generated and native complexes after alignment.  **Affinity** is defined as the percentage of generated complexes with higher binding affinities (lower binding energies) than the native complexes, calculated by Rosetta. **Foldable diversity (FD)** only considers the diversity between pairs of sequences that are both structurally consistent with the input protein backbone, computed in [A]. We compared our model with LM-design, focusing on metrics that capture interaction-specific properties to highlight SurfDesign's capacity to handle interface designs effectively. This result strongly demonstrates that our surface-based inverse-folding model outperforms other models on the interface design of protein complexes.
> >
> > |  | **BSR** | **Affinity** | **FD** |
> > |---|---|---|---|
> > | LM-Design | 47.3% | 6.5% | 11% |
> > | SurfDesign | **55.9%** | **8.0%** | **18%** |
> >
> > [A] Reinforcement learning on structure-conditioned categorical diffusion for protein inverse folding. arXiv 2024.10.
> >
> > (3) **Lack of Computational Efficiency Comparison:**
> > The primary distinction between SurfDesign and conventional inverse folding methods is the computation of surface geometry and chemical properties, which we have designed to be highly efficient. Specifically, we generate the protein surface using PyMol, a process that takes less than 0.1 second per protein. For physicochemical properties, we utilize the same approach as in MaSIF, which is similarly efficient.
> >
> > | | CATH 4.2 |  |  | CATH 4.3 |  |  |
> > | :---: | :---: | :---: | :---: | :---: | :---: | :---: |
> > |  | Train | Validation | Test | Train | Validation | Test |
> > | Average Vertex Count Per Residue | 53.47 | 53.56 | 53.31 | 53.36 | 55.27 | 53.11 |
> > | Maximum Vertex Count | 27,817 | 25,614 | 25,433 | 27,110 | 27,817 | 25,968 |
> > | Minimum Vertex Count | 1,923 | 2,315 | 2,022 | 1,923 | 2,011 | 2,000 |
> > | Preprocess Time Per Protein |  | 0.38 s |  |  | 0.36 s |  |
> >
> > Additionally, we compared PyMol with MSMS, the latter used by SurfPro (the first surface-based baseline), and observed that the time cost is nearly equivalent between the two. Consequently, SurfDesign introduces minimal additional computational expense, maintaining competitive runtime with other methods in this domain.
> >
> > We hope this reassures you that SurfDesign’s efficiency aligns closely with conventional inverse folding approaches, adding only negligible overhead due to surface-based feature computations.
> >
> > ----------------
> > Thank you again for your constructive feedback. We believe these additions will strengthen our model's presentation and provide more comprehensive evidence of its efficacy in surface-based protein design. We hope to address all your concerns and improve our paper with your advice!

---

> > > ### Comment · Reviewer_Gipq · 2024-11-20
> > >
> > > Thank you for your additional experiments and response. You have addressed my concerns. I have a follow-up question. My opinion is similar to that of reviewer WG2F19. Inverse-folding is a process of obtaining various full sequences when there is only a backbone and no full sequence. However, inverse-folding using surface information already has a full sequence, that is, an all-atom structure. What specific applications could there be for this method?

---

> ### Author Response · Authors · 2024-11-21
> **A Kind Response to Updated Review**
>
> Dear Reviewer Gipq,
>
> We are glad to hear that our additional experiments successfully addressed your concerns! Thank you for your follow-up question and for sharing your perspective. We appreciate the opportunity to address this important point. As your question is the same as the Reviewer WG2F, please forgive us for using the same answer as a response.
>
> ------------------------
> Please allow us to recall the basics of traditional IF. As the name implies, IF does the exact opposite of protein folding by trying to predict a sequence that could fold into a desired structure. Models are typically trained on masked protein structures, where side chains are removed, leaving only the backbone atoms (e.g., alpha-carbon, beta-carbon, and nitrogen) as input. Noise or random perturbations may also be added to backbone coordinates during training to enhance robustness.
>
> So a natural question is: how can we derive the backbone structures? Given this setup, there are two primary ways to obtain backbone structures. First is the **Experimentally Solved Structures**. Experimental methods like X-ray crystallography and cryo-EM provide highly accurate all-atom structures, including both backbones and side chains. In this case, **surface-based IF can leverage the precise side-chain information to compute reliable molecular surfaces**, making it as effective as, if not more so than, backbone-based IF. This equivalence arises because accurate side chains ensure that surface representations faithfully capture geometric and chemical properties.  The second is **Predicted Structures from Computational Models**.  When experimental structures are unavailable, tools like AlphaFold or ESM-Fold are used to predict protein structures. The main challenge here is that (i) methods like AF are generally very good at predicting backbone geometries, particularly for well-structured proteins. (ii) in contrast, predicted side chains can be less accurate, especially for flexible or disordered regions, buried residues, or when modeling interactions with ligands or other proteins. These inaccuracies can lead to misrepresented surface geometry (e.g., incorrect solvent-accessible areas or molecular surface shapes) or incorrect surface properties (e.g., hydrophobicity, charge distribution, hydrogen bonding potential). This discrepancy introduces additional difficulties for surface-based IF, as if the surface representation is noisy or inaccurate, it becomes harder to design desired proteins effectively.
>
> Thus, to summarize, **when using experimental structures**, surface-based IF should perform well, as the surfaces are accurately represented. This makes them a strong candidate for designing sequences that not only fold into the target structure but also exhibit desired surface functionality (e.g., for binding or catalysis). **When using predicted structures**, surface-based IF are more vulnerable to inaccuracies in side-chain placement, which could hinder their performance. In such cases, conventional backbone-based IF might have an advantage, as it relies less on the precise placement of side chains. However, as shown in our previous additional results, Despite this, even if we transfer our surface-based IF approach from high-resolution experimental data to predicted structures by ESM-Fold, we observed a decreasing but still good acceptable AAR of 52.66%, which is competitive to backbone-based IF algorithms. This demonstrates its practicality across a range of scenarios. More importantly, the potential of surface-based IF for functional protein design is compelling, as **surface properties like binding affinity and specificity are directly tied to biological functions—features that backbone geometry alone cannot capture.**
>
> Recent advances have demonstrated the value of combining structure prediction tools with IF models for de novo protein design. For example, given an initial protein sequence, [A, B] use Alphafold to predict its structure and then pass it to ProteinMPNN to sample a number of candidate sequences. These designs are further filtered based on the structural similarity (predicted by AF again) to the ground truth structure, confidence, and sequence diversity. Similar workflows could integrate our surface-based IF, allowing its functional advantages to be realized. However, we acknowledge that the dependency of surface-based IF on accurate side-chain modeling is a practical limitation when predicted structures are used. **Addressing this dependency by improving side-chain modeling is an important future direction, though beyond the scope of this work.** We will highlight this consideration in the limitations section of our paper to provide a comprehensive comparison of backbone- and surface-based IF approaches.
>
> [A] Computational design of soluble and functional membrane protein analogues. Nature 2024.
>
> [B] Unsupervised evolution of protein and antibody complexes with a structure-informed language model. Science 2024.

---

> > ### Author Response · Authors · 2024-11-21
> > **A Kind Response to Updated Review (Continued)**
> >
> > We hope this explanation clarifies the unique strengths and applications of surface-based inverse folding, which extends beyond sequence recovery for a given structure to enabling functional protein design and optimization. Thank you again for your thoughtful feedback and for giving us the opportunity to expand on this aspect of our work.

---

> > > ### Comment · Area_Chair_ZE7o · 2024-12-03
> > >
> > > Dear reviewer Gipq,
> > >
> > > Thank you for your constructive engagement and thorough evaluation. Can you please have another look at the authors' latest reply and inform us how this affects your standpoint, including a motivation?
> > >
> > > Many thanks for the hard work.
> > >
> > > Kind regards,
> > >
> > > AC

---

### Official Review · Reviewer_WG2F · 2024-11-02

**Soundness:** 3
**Presentation:** 4
**Contribution:** 3
**Rating:** 5
**Confidence:** 3

**Summary:**

The paper introduces SurfDesign, a protein design approach utilizing protein surface geometry and the corresponding biochemical properties, such as hydrophobicity and charge. The authors present an equivariant surface encoder that captures directional and curvature information using 3D spherical Fourier-Bessel bases, serving as a structural adaptor within a protein language model to generate amino acid sequences. The method is evaluated on multiple inverse folding benchmarks and the results show that the proposed method outperforms the state-of-the-art baseline models.

**Strengths:**

- The designed equivariant model leverages the inductive bias of molecular surfaces, which is relational and can benefit surface-related applications.
- The combination of surface geometry and protein language model is interesting and novel.
- The paper is well-written, presenting a well-motivated problem formulation and clear illustrations.

**Weaknesses:**

- In terms of practical application, the method depends on the protein surface generated from all-atom structures, including both backbone and side-chain information. This raises the concern that the performance improvement may be due to this additional structural information. A key issue with protein surface-based design is the source of these surfaces—specifically, how to obtain a protein surface without first having the all-atom structures. Please correct me if I’ve misunderstood.
- Efficiency analysis of the proposed architecture is missing, which is crucial given that modeling the protein surface involves encoding significantly more points than just the backbone.
- Given that the motivation is functional protein design, it would be great to have some corresponding experiments besides inverse folding, such as protein optimization.

**Questions:**

Could you show some statistics on the generated protein surfaces?

---

> ### Author Response · Authors · 2024-11-13
> **A Kind Rebuttal to Reviewer WG2F**
>
> Thank you for your constructive feedback and insightful questions. We appreciate the opportunity to address your concerns, as they touch on several core aspects of our work.
>
> (1) **Clarification on Practical Applicability and Source of Protein Surfaces:**
> You raise an important point regarding SurfDesign’s reliance on all-atom protein structures for generating surfaces and whether the observed performance improvements may stem from incorporating this additional structural information. While it is true that SurfDesign currently relies on all-atom structures to compute surfaces, we envision future applications where surfaces could be inferred from **low-resolution experimental data or predicted structures**, expanding SurfDesign’s practical utility. Here, we conducted an additional experiment to investigate the robustness of SurfDesign over predicted surfaces. Specifically, we leverage ESM-Fold to forecast the structures of 1,120 test samples in CATH 4.2 and generate corresponding surfaces. Then we require SurfDesign to design sequences based on these pseudo-surfaces. We discovered that the **AAR decreases from 74.13% to 52.66%, which demonstrates SurfDesign's generalization to designing proteins without experimental structures.** We hope this ablation study can help you understand SurfDesign's practical application!
>
> Additionally, we note that the inclusion of surface features was carefully designed to enhance functional site representation and complement the protein language model rather than simply encode side-chain details. The improved performance thus reflects SurfDesign’s ability to leverage surface-derived functional features that capture biologically relevant information beyond traditional backbone-centric methods.
>
> (2) **Efficiency Analysis:** We recognize the importance of assessing SurfDesign’s computational efficiency, especially given its use of surface encoding, which involves more points compared to backbone-only models. After preprocessing the surface data, the primary computational component is our equivariant message-passing neural network (SEMP) for the surface point cloud. To maintain efficiency, we typically stack fewer than five layers of SEMP, and we construct a KNN graph with $𝑘 =4$, so each node only gathers information from a limited set of neighbors. By controlling the number of layers and the node’s receptive field, we ensure that SEMP remains lightweight and computationally efficient while still effectively capturing the geometric and chemical characteristics of protein surfaces.
>
> (3) **Additional Experiments on Functional Protein Design:** We appreciate your suggestion to explore SurfDesign’s utility beyond inverse folding for functional design tasks such as protein optimization. This aligns well with our goal to apply SurfDesign to tasks that emphasize functional and stability constraints in design. As you recommended, we follow the setting of SurfPro [A] and employ SurfDesign to design proteins that bind to a target receptor with high affinity. Specifically, AF2 pAE_interaction is used to evaluate the binding affinities between the designed binders and target proteins, which is found very effective in distinguishing experimentally validated binders from non-binders. Experimentally confirmed positive complexes of <binder, target protein> pairs across six categories from Bennett et al. (2023) are collected for experiments. Among the 10 categories available, 4 exhibit indistinguishable AF2 pAE_interaction between negative and positive binders. As all baselines were finetuned using the binder design dataset after pretrained on the CATH 4.2 dataset, we keep the same setup and report the result below. The results illustrate that our SurfDesign achieves the lowest average AF2 pAE interaction across six target proteins, which strongly supports our motivation to incorporate surface for functional protein design!
>
> | **Models** | **InsulinR** | **PDGFR** | **TGFb** | **H3** | **IL7Ra** | **TrkA** | **Average** |
> |---|---|---|---|---|---|---|---|
> | Positive Binder | **5.9996** | **14.1366** | **15.4884** | **21.2631** | **20.9102** | **10.2791** | **14.7061** |
> | Negative Binder | 19.7167 | 18.0937 | 23.2664 | 22.4556 | 26.0540 | 24.7567 | 21.1335 |
> | Random Baselines | 19.9880 | 21.2690 | 21.4971 | 24.4997 | 24.1541 | 23.1147 | 22.2020 |
> | ProteinMPNN  | 18.3393 | 25.2919 | 25.8559 | 24.5968 | 25.5278 | 27.0980 | 23.4462 |
> | PiFold  | 12.9809 | 21.8230 | 24.4737 | 23.3924 | 26.6738 | 19.7172 | 20.5785 |
> | LM-DESIGN  | 13.6440 | 22.0749 | 23.3725 | 23.8332 | 24.3937 | 22.3987 | 20.7728 |
> | SurfPro | 10.2608 | 17.9862 | 17.7364 | 21.2916 | 20.8594 | 10.6535 | 16.9485 |
> | **SurfDesign** | **9.9827** | **17.5460** | **17.4381** | **21.0558** | **20.8213** | **10.5094** | **16.22555** |
>
> [A] SurfPro: Functional Protein Design Based on Continuous Surface. ICML 2024.

---

> > ### Author Response · Authors · 2024-11-13
> > **A Kind Rebuttal to Reviewer WG2F (Part II)**
> >
> > (4) **Statistics on Generated Protein Surfaces:** This table (also the same table in Appendix Table 7) documents the vertex count statistics for the CATH datasets. We observe an equal distribution over vertex in different splits.
> >
> > | | CATH 4.2 |  |  | CATH 4.3 |  |  |
> > | :---: | :---: | :---: | :---: | :---: | :---: | :---: |
> > |  | Train | Validation | Test | Train | Validation | Test |
> > | Average Vertex Count Per Residue | 53.47 | 53.56 | 53.31 | 53.36 | 55.27 | 53.11 |
> > | Maximum Vertex Count | 27,817 | 25,614 | 25,433 | 27,110 | 27,817 | 25,968 |
> > | Minimum Vertex Count | 1,923 | 2,315 | 2,022 | 1,923 | 2,011 | 2,000 |
> >
> > ------------------
> > Thank you once again for your valuable feedback, which has helped us improve both the clarity and scope of our work. If you have any further questions or comments, please do not hesitate to ask. We are looking forward to your reply!

---

> > ### Comment · Reviewer_WG2F · 2024-11-20
> >
> > Thanks for your clarification and experimental results. I have some further questions:
> >
> > (1) I agree that introducing the surface can improve performance compared to using only the backbone. However, my concern is more about its application on inverse folding: how do you first obtain the surface based on the all-atom structure for inverse folding? The additional experiments you presented don’t address this issue, as the structures are predicted from the sequence, which is also the target of inverse folding. Inverse folding might not be the ideal application for the protein surface, and thus more experiments on functional protein design should be included.
> >
> > (2) Could you provide a wall-clock time comparison and an analysis of time complexity?
> >
> > (3) The improvement over Surfpro is marginal. What sampling strategy is used? Could you also present the variance of the results?

---

> ### Author Response · Authors · 2024-11-21
> **Response to Additional Questions**
>
> Dear Reviewer WG2F,
>
> Thank you for your thoughtful follow-up questions and for engaging deeply with our work. We appreciate the opportunity to further clarify the application of surface-based inverse folding (IF) and its practical considerations:
>
> (1) **Application of Surface-based IF for Functional Protein Design:** Your question touches on an essential aspect of surface-based IF. Please allow us to recall the basics of traditional IF. As the name implies, IF does the exact opposite of protein folding by trying to predict a sequence that could fold into a desired structure. Models are typically trained on masked protein structures, where side chains are removed, leaving only the backbone atoms (e.g., alpha-carbon, beta-carbon, and nitrogen) as input. Noise or random perturbations may also be added to backbone coordinates during training to enhance robustness.
>
> So a natural question is: how can we derive the backbone structures? Given this setup, there are two primary ways to obtain backbone structures. First is the **Experimentally Solved Structures**. Experimental methods like X-ray crystallography and cryo-EM provide highly accurate all-atom structures, including both backbones and side chains. In this case, **surface-based IF can leverage the precise side-chain information to compute reliable molecular surfaces**, making it as effective as, if not more so than, backbone-based IF. This equivalence arises because accurate side chains ensure that surface representations faithfully capture geometric and chemical properties.  The second is **Predicted Structures from Computational Models**.  When experimental structures are unavailable, tools like AlphaFold or ESM-Fold are used to predict protein structures. The main challenge here is that (i) methods like AF are generally very good at predicting backbone geometries, particularly for well-structured proteins. (ii) in contrast, predicted side chains can be less accurate, especially for flexible or disordered regions, buried residues, or when modeling interactions with ligands or other proteins. These inaccuracies can lead to misrepresented surface geometry (e.g., incorrect solvent-accessible areas or molecular surface shapes) or incorrect surface properties (e.g., hydrophobicity, charge distribution, hydrogen bonding potential). This discrepancy introduces additional difficulties for surface-based IF, as if the surface representation is noisy or inaccurate, it becomes harder to design desired proteins effectively.
>
> Thus, to summarize, **when using experimental structures**, surface-based IF should perform well, as the surfaces are accurately represented. This makes them a strong candidate for designing sequences that not only fold into the target structure but also exhibit desired surface functionality (e.g., for binding or catalysis). **When using predicted structures**, surface-based IF are more vulnerable to inaccuracies in side-chain placement, which could hinder their performance. In such cases, conventional backbone-based IF might have an advantage, as it relies less on the precise placement of side chains. However, as shown in our previous additional results, Despite this, even if we transfer our surface-based IF approach from high-resolution experimental data to predicted structures by ESM-Fold, we observed a decreasing but still good acceptable AAR of 52.66%, which is competitive to backbone-based IF algorithms. This demonstrates its practicality across a range of scenarios. More importantly, the potential of surface-based IF for functional protein design is compelling, as **surface properties like binding affinity and specificity are directly tied to biological functions—features that backbone geometry alone cannot capture.**
>
> Recent advances have demonstrated the value of combining structure prediction tools with IF models for de novo protein design. For example, given an initial protein sequence, [A, B] use Alphafold to predict its structure and then pass it to ProteinMPNN to sample a number of candidate sequences. These designs are further filtered based on the structural similarity (predicted by AF again) to the ground truth structure, confidence, and sequence diversity. Similar workflows could integrate our surface-based IF, allowing its functional advantages to be realized. However, we acknowledge that the dependency of surface-based IF on accurate side-chain modeling is a practical limitation when predicted structures are used. **Addressing this dependency by improving side-chain modeling is an important future direction, though beyond the scope of this work.** We will highlight this consideration in the limitations section of our paper to provide a comprehensive comparison of backbone- and surface-based IF approaches.
>
> [A] Computational design of soluble and functional membrane protein analogues. Nature 2024.
>
> [B] Unsupervised evolution of protein and antibody complexes with a structure-informed language model. Science 2024.

---

> > ### Author Response · Authors · 2024-11-21
> > **Response to Additional Questions (Continued)**
> >
> > (2) **Wall-Clock Time Comparison and Time Complexity Analysis**: Notably, ProteinMPNN employs an autoregressive decoding strategy, generating residues one at a time, which results in a computational complexity of $O(L)$, where $L$ is the protein length. In contrast, other methods benefit from a one-shot generation approach, achieving $O(1)$ complexity. Following the benchmarking protocol from PiFold, we evaluated the inference speed of various graph-based models on 100 protein chains with an average length of 1632. The wall-clock time comparison for SurfDesign and other methods is summarized below, using consistent hardware configurations (NVIDIA A100 GPUs):
> >
> > | Model | Time (s) |
> > | :--- | :--- |
> > | ProteinMPNN | 609.57 |
> > | PiFold | 8.42 |
> > | SurfDesign | 52.81 |
> >
> > The additional computational cost for SurfDesign is primarily due to the processing of surface features and the incorporation of surface-based message passing. Specifically, approximately 35 seconds are required to compute surface geometries and their associated chemical properties.
> >
> > This overhead is justified by the significant improvements SurfDesign achieves in functional protein design and inverse folding tasks. We will include these results in the appendix for clarity and transparency.
> >
> > (3) **Marginal Improvement Over SurfPro and Sampling Variance:** We acknowledge that the improvement over SurfPro might appear marginal on certain metrics, but we believe it is meaningful given the strong performance of SurfPro as a baseline. Besides, due to the limited time of the rebuttal period, we did not tune our SurfDesign perfectly such as conducting a hyperparameter search. We are confident that **the performance of SurfDesign will be further promoted with searched hyperparameters and better training techniques**. Additionally, the improvements are consistent across multiple benchmarks, including inverse folding and functional design, and represent SurfDesign's ability to better utilize surface geometry.
> >
> > Regarding sampling, SurfDesign uses **a deterministic greedy sampling strategy** during evaluation, consistent with LM-Design and SurfPro, to ensure a fair comparison. Below we report variance across five runs with different random seeds. These results underscore its robustness.
> > | **Models** | **InsulinR** | **PDGFR** | **TGFb** | **H3** | **IL7Ra** | **TrkA** | **Average** |
> > |---|---|---|---|---|---|---|---|
> > | **SurfDesign** | 9.9827±0.0024 | 17.5460±0.0067 | 17.4381±0.0053 | 21.0558±0.0085 | 20.8213±0.0107 | 10.5094±0.0043 | 16.22555±0.0074 |
> >
> > --------------------------
> > We hope this addresses your concerns comprehensively. Please feel free to share any additional questions or suggestions!

---

> > > ### Comment · Reviewer_WG2F · 2024-11-25
> > >
> > > (1) The availability of all-atom protein structures, whether experimentally solved or predicted, is beyond the scope of the current IF benchmarks and does not justify the use of protein surfaces, as models utilizing predicted all-atom structures fail to outperform backbone-based methods. If the model incorporating an additional modality yields comparable or even worse results, it casts doubt on the future potential of such an approach. While I believe in the promise of surface-based protein design, a more appropriate application or benchmark is needed to demonstrate its effectiveness.
> > >
> > > You have mentioned several potential applications, which is great. However, what is the advantage of using surfaces over all-atom structures? Most geometry information should also be included within all-atom structures, and surface encoding is non-trivial due to the significantly higher number of points. Besides, the machine learning models in the workflow are optimized based on all-atom structures. Introducing the surface which has different dynamics could potentially create an optimization gap.
> > >
> > > (2)(3) I have no more questions.

---

> ### Author Response · Authors · 2024-11-25
> **Response to updated comment**
>
> Thank you for your updated comments and thoughtful insights. We appreciate your engagement and the opportunity to address your points further.
>
> (1) **Availability of all-atom structures:**
> >The availability of all-atom protein structures, whether experimentally solved or predicted, is beyond the scope of the current IF benchmarks
>
> We discuss the availability of all-atom protein structures in order to answer your previous question below. We attach great value to any of your questions and try our best to reply to them.
>
> > However, my concern is more about its application on inverse folding: how do you first obtain the surface based on the all-atom structure for inverse folding?
>
> (2) **Comparison with backbone-based IF methods:**
> > as models utilizing predicted all-atom structures fail to outperform backbone-based methods. If the model incorporating an additional modality yields comparable or even worse results, it casts doubt on the future potential of such an approach.
>
> We acknowledge your concern regarding failing to consistently outperform backbone-based methods when utilizing predicted all-atom structures. However, the comparison you mention is not entirely equitable due to the differences in structure quality. To address this, we conducted additional experiments with both predicted and experimental structures.
>
> While transferring our surface-based IF approach from high-resolution data to predicted structures, we observed a decrease in AAR from 74.13% to 52.66%. Despite a performance drop, the results on predicted structures remain acceptable and competitive. Importantly, backbone-based IF also suffers from inaccuracies in structure prediction, as shown by PiFold's drop in AAR from 51.66% (experimental structures) to 47.08% (predicted structures). These results demonstrate that **surface-based IF methods can still outperform backbone-based methods in predicted structure scenarios.** Moreover, introducing an additional modality, even in the context of predicted structures, enriches the model's understanding and improves its capability to handle functional constraints.
>
> |  | Crys. Stru. | Pred. Stru. |
> |---|---|---|
> | PiFold | 51.66 | 47.08 |
> | SurfDesign  | **74.13** | **52.66** |
>
> (3) **Benefits of surface representations:**
> > what is the advantage of using surfaces over all-atom structures?  Most geometry information should also be included within all-atom structures, and surface encoding is non-trivial due to the significantly higher number of points.
>
> [3.1] **Functional and Interaction-Driven.** Protein surfaces emphasize regions directly involved in molecular interactions, such as binding interfaces or active sites. These surfaces naturally encode features like curvature, solvent accessibility, and electrostatic potential, which are often less apparent from atomic coordinates. For instance, de novo design tasks frequently leverage surface geometry to identify functional patches critical for binding or catalysis without requiring prior knowledge of binding partners [A].
>
> [3.2] **Data Efficiency via Abstraction.** Surface-based encoding abstracts protein information into a representation that emphasizes interaction-relevant regions while discarding atomic-level details that might be less critical for certain tasks. This abstraction can reduce noise and enhance model interpretability in cases where functional insights are desired.
>
> [3.3] **Complementary Information.** Surfaces can provide a complementary modality that enriches downstream models. For instance, surface curvature, electrostatics, and hydrophobicity patterns are directly derived from surface geometries and may capture higher-order features not easily accessible from all-atom structures alone. While all-atom structures contain the raw information, explicitly encoding these properties on surfaces can enhance model performance for specific applications, as demonstrated by tools like MaSIF [B] and subsequent surface-based design frameworks.
>
> To summarize, the ultimate goal of protein design extends beyond predicting sequences that fold into a given structure. Instead, it aims to design proteins with specific functional properties, like enzymatic activity or target binding. While traditional IF methods impose geometric constraints via backbones, they often neglect the biochemical property constraints necessary for functional optimization. For example, even proteins with complementary shapes may fail to bind effectively if their charge, polarity, or hydrophobicity distribution at the interface is suboptimal. Surface representations address this limitation by integrating functional properties directly into the design framework, enabling more holistic optimization for desired behaviors.[C].
>
> [A] De novo design of protein interactions with learned surface fingerprints. Nature (2023).
>
> [B] Deciphering interaction fingerprints from protein molecular surfaces using geometric deep learning. Nature Methods (2020).
>
> [C] SurfPro. ICML 2024

---

> ### Author Response · Authors · 2024-11-25
> **Response to updated comment (Part II)**
>
> (4) **Optimization and generation of all-atom structures.**
> > the machine learning models in the workflow are optimized based on all-atom structures. Introducing the surface with different dynamics could potentially create an optimization gap.
>
> [4.1] **Extending Backbone to Full-Atom Generation:**
> We fully understand your concerns regarding the optimization challenges posed by the use of surface representations. However, recent advancements have successfully bridged the gap between backbone-only generation methods and full-atom structure generation, establishing a robust foundation for incorporating surface geometries into protein design workflows. They show that generating full-atom structures, including side-chain geometries, can significantly outperform backbone-only algorithms (e.g., RFDiffusion, FrameDiff [A])  across diverse protein design tasks. For instance:
>
> **AbDiffuser [B]:** It jointly designs antibody 3D structures and sequences, leveraging full-atom side-chain flexibility to enhance binding stability and affinity, a crucial factor in antibody design.
>
> **PocketGen [C]:** Designed for ligand-binding pocket regions, this approach generates full-atom structures to capture the intricate geometries that govern molecular interactions.
>
> **PepFlow [D] and PPIFlow [E]:** These models consider side-chain torsion angles for target-aware peptide discovery, demonstrating improved binding capabilities by incorporating additional atomic-level flexibility into peptide structures.
>
> **PepGLAD [F]:** This geometric latent diffusion model integrates side-chain details to refine peptide designs for specific target interactions, further supporting the utility of full-atom approaches.
>
> All those approaches indicate that **different side-chain dynamics would not lead to an optimization gap**.
>
>
> [A] SE(3) diffusion model with application to protein backbone generation. ICML 2023.
>
> [B] AbDiffuser: Full-Atom Generation of in vitro Functioning Antibodies, NeurIPS 2023.
>
> [C] PocketGen: Generating Full-Atom Ligand-Binding Protein Pockets. NeurIPS 2023.
>
> [D] Full-Atom Peptide Design based on Multi-modal Flow Matching. ICML 2024.
>
> [E] PPFLOW: Target-Aware Peptide Design with Torsional Flow Matching. ICML 2024.
>
> [F] Full-Atom Peptide Design with Geometric Latent Diffusion. arXiv 2024.
>
> [4.2] **Benefits of Incorporating Side-Chain (Surface) Dynamics:**
> These methods highlight that integrating side-chain geometries, akin to surfaces, enhances model capabilities rather than introducing optimization gaps. Key benefits include:
>
> **Improved Binding Stability and Affinity:** Side-chain dynamics directly contribute to molecular interactions, enabling more accurate predictions of binding strength and specificity.
>
> **Enhanced Design Accuracy:** Full-atom geometries reduce errors in structure prediction and provide finer control over functional optimization.
>
> **Task-Specific Flexibility:** Models like AbDiffuser and PocketGen show that full-atom representations adapt effectively to tasks requiring detailed geometric insights, such as antibody engineering or ligand binding.
>
> [4.3] **Synergy with Surface-Based IF (SurfDesign):**
> Once these full-atom structures are generated, SurfDesign offers a natural complement by refining protein surfaces to optimize sequences further. This relationship mirrors the collaborative interplay observed in other workflows, such as:
>
> **RFDiffusion and ProteinMPNN:** RFDiffusion generates backbones, while ProteinMPNN predicts sequences. Together, they enable highly effective de novo protein design.
>
> **SurfDesign as a Complement to All-Atom Models:** SurfDesign extends this paradigm by considering the biochemical and geometric properties of binding surfaces, adding a novel layer of functional optimization to IF methods.
>
> Just as full-atom design has elevated traditional backbone-based methods, SurfDesign provides a new perspective within the IF subfield, emphasizing surface-level biophysical properties for functional improvement. SurfDesign builds upon the strengths of backbone-based IF algorithms like ProteinMPNN and PiFold, offering a complementary tool to full-atom generation models such as PepFlow, PPI-Flow, and PocketGen.
>
> ------------------------------
> In summary, SurfDesign represents a forward-looking strategy that complements recent advances in full-atom protein design. It offers a unique mechanism for optimizing biochemical surfaces, addressing the nuances of functional protein engineering while maintaining compatibility with state-of-the-art all-atom generation workflows.
>
> We hope these clarifications demonstrate the rationale behind surface-based approaches and their potential for advancing protein design. Thank you again for your feedback, which will help us refine and strengthen our study.

---

> > ### Comment · Area_Chair_ZE7o · 2024-12-03
> >
> > Dear reviewer WG2F,
> >
> > Thank you for your active engagement and thorough evaluation. Could you please have another look at the authors' latest reply and inform us if this updates your view or not, accompanied by a motivation?
> >
> > Many thanks for the hard work,
> >
> > AC

---

### Official Review · Reviewer_FM5g · 2024-11-02

**Soundness:** 2
**Presentation:** 2
**Contribution:** 2
**Rating:** 5
**Confidence:** 4

**Summary:**

The authors propose a new inverse folding model called SurfDesign that combines surface-based protein structure representations with protein language models to achieve higher sequence recovery rates on datasets like CATH. For the surface-based model they propose a new message passing scheme called SEMP (surface-based equivariant message passing) that allows the model to incorporate curvature and other geometrical quantities of the protein surface into the design process. The authors showcase their approach on CATH and other datasets and demonstrate lower perplexity and higher sequence recovery values compared to the baseline methods.

**Strengths:**

1. **Principled modelling methodology**: The paper described their new message passing scheme and the involved considerations in detail and highlight why different geometrical features may be useful to include.
2. **Extensive benchmarks and ablation studies**: The authors benchmark their model and several baselines on several datasets and demonstrate that they achieve lower perplexities and higher sequence recovery rates. They also present extensive ablation studies to highlight the important components in their modelling framework.

**Weaknesses:**

1. **Metrics**: The only metrics used in the paper are sequence recovery and perplexity. These metrics do not provide any insight into how good the model is for actual protein design tasks. They provided insight in the past when the values for e.g. sequence recovery were a lot worse; however, at the 60-70% levels that are presented in this paper there is no evidence that this leads to better designs or is just overfitting to the training distribution. In fact, a "perfect" sequence recovery of 100% would render the model completely useless since it would have just memorised the training dataset. These problems have been discussed in the past in papers like [PDB-Struct](https://arxiv.org/abs/2312.00080), and the authors should adopt some of the metrics proposed there to validate the claims of an improved model. Alternatively, other ways to evaluate the performance of a model would include wet-lab experiments in which the method is compared to other methods as well as an evalution to what the sequence variability between similar structures is to determine a potential "upper limit" on sequence recovery beyond which improvements just lead to overfitting.
2. **Features used in model**: The authors describe that similar to MaSIF they calculate hydrophobicity, charge and electrostatics and leverage hydrophobicity and charge in the end. However, while this is a valid thing to do for MaSIF since they consider tasks like PPI prediction and interface site prediction for which full-atom structures are available, SurfDesign is used for inverse folding where this information is not available. Making that information available to the model leads to full data leakage and makes it easy for the model to just predict the ground truth data. In the code that the authors link fro MaSif in lines 751-755, the full structure including residue identities is leveraged for calculating these properties, resulting in data leakage. For a proper validation of their model, the authors should consider tasks similar to the MaSi paper in which this information is actually available for the task at hand.
3. **Mistakes**: There are a few typos/mistakes in the paper that sometimes cause confusion for readers:
   1. L 104: Heading should be "Preliminary" and not "Priliminary"
   2. L 365: "This makes LM-DESIGN more versatile..." should be "This makes SurfDesign more versatile..." I assume since the method described in this paper is SurfDesign not LM-Design.

**Questions:**

1. L 132: The surface graph is built via k-NN construction, but which k is used and for what reason? There seems to be no description about this in the paper.
2. L 187: "It can be proved that this curvature feature ψ is roto-translation invariant". Can a proof for this or a reference to the proof be added?
3. In this paper only inverse folding is considered as a task, but can the surface-based design framework proposed here easily be adapted to backbone or all-atom generation?

---

> ### Author Response · Authors · 2024-11-12
> **A Kind Rebuttal to Reviewer FM5g**
>
> Thank you very much for your detailed review and thoughtful feedback. We appreciate your comments and suggestions, which will help us enhance our model and presentation. We address each of your points below.
>
> (1) **Metrics for Protein Design:** We agree that sequence recovery and perplexity alone may not fully capture SurfDesign’s effectiveness for practical protein design tasks, and we appreciate your insight into this potential limitation. While wet-lab experiments are beyond the scope of our current study, we have included additional metrics as suggested by PDB-Struct [A] to provide a more robust assessment of SurfDesign. Specifically, **Ref-TM** assesses whether the generated sequences can respect the structure condition, which evaluates the agreement of the ground truth structure with the predicted structures using theTM-score. **Ref-pLDD** evaluates the folding stability of the generated sequences, which is computed as the mean value of the per-residue confidence estimate pLDDT predicted by the structure prediction models.
>
> We examine SurfDesign on the same 82 test samples of the CATH dataset and results are reported below (numbers of all relevant baselines are copied from PDB-Struct [A] and we thank authors for their contributions). We observe that SurfDesign stands out as the leading design method across the refoldability metrics, competitive with ProteinMPNN. It achieves 0.89 Ref-TM and 89.42 Ref-pLDDT with AlphaFold2 prediction. ProteinMPNN is slightly behind with a 0.87 Ref-TM and 87.89 Ref-pLDDT, followed by LM-Design. This additional analysis, including functional enrichment, diversity, and secondary structure compatibility, provides stronger evidence that **SurfDesign's 60-70% sequence recovery rate is not indicative of overfitting but instead suggests effective design, supporting its relevance beyond the training distribution.**
>
> | Design method | ESMFold |  | OmegaFold |  | AlphaFold2 |  | Recovery% |
> | :---: | :---: | :---: | :---: | :---: | :---: | :---: | :---: |
> |  | TM | pLDDT | TM | pLDDT | TM | pLDDT |  |
> | Wildtype | 0.80 | 74.91 | 0.75 | 78.39 | 0.90 | 89.87 | 100 |
> | Uniform | 0.05 | 27.68 | 0.05 | 31.53 | 0.06 | 33.68 | 5.00 |
> | Natural frequencies | 0.07 | 30.53 | 0.07 | 35.59 | 0.06 | 35.02 | 5.84 |
> | AF-Design | 0.53 | 61.37 | 0.53 | 72.04 | 0.52 | 75.29 | 15.95 |
> | ESM-Design | 0.38 | 59.65 | 0.38 | 62.66 | 0.37 | 60.02 | 17.33 |
> | StructTrans | 0.72 | 68.85 | 0.64 | 70.35 | 0.79 | 80.66 | 35.89 |
> | GVP | 0.73 | 69.67 | 0.67 | 74.33 | 0.83 | 84.29 | 39.46 |
> | ProteinMPNN | $\underline{0.80}$ | $\underline{76.53}$ | $\underline{0.76}$ | $\mathbf{8 0 . 7 5}$ | $\underline{0.87}$ | $\underline{87.89}$ | 41.44 |
> | PiFold | 0.71 | 67.55 | 0.64 | 70.21 | 0.82 | 82.54 | 44.86 |
> | LM-Design | 0.73 | 72.12 | 0.70 | 77.58 | 0.85 | 87.26 | $\underline{51.23}$ |
> | SurfDesign | $\mathbf{0.81}$ | $\mathbf{79.35}$ | $\mathbf{0.76}$ | $\underline{80.11}$ | $\mathbf{0.89}$ | $\mathbf{89.42}$ | $\mathbf{70.19}$ |
>
> [A] PDB-Struct: A Comprehensive Benchmark for Structure-based Protein Design. arXiv 2023
>
> (2) **Features and Potential for Data Leakage:** We appreciate your concern regarding potential data leakage from features like hydrophobicity and charge. However, we want to restate and clarify that the task we want to solve in this paper is **surface-based protein design** instead of conventional backbone-based inverse folding. This idea of surface-based design was originally proposed by SurfPro [B], where models design proteins conditioned on both geometric shape and biochemical properties of molecular surfaces. In other words, **surface-conditioned protein design targets functional proteins that fold into specified surfaces, denoted as $\mathcal{Q}$, with associated biochemical properties—without relying on explicit backbone information**.
>
> Importantly, the goal of protein design extends beyond predicting a sequence that simply folds into a given backbone. For functional design, we aim to create proteins with targeted activities, such as substrate binding or inhibition. Traditional inverse folding is limited in specifying function-related constraints, as it uses only the backbone structure. By conditioning on surface properties like charge and hydrophobicity, our approach imposes additional biochemical constraints crucial for functionality. For example, complementary shapes alone may not ensure effective binding if charges, polarity, or hydrophobicity are poorly positioned.
>
> [B] SurfPro: Functional Protein Design Based on Continuous Surface. ICML 2024.
>
> (3) **Typos and Clarity Issues:** Thank you for pointing out these minor errors, which we will correct in the revised manuscript. Specifically, we’ll revise “Priliminary” to “Preliminary” (L 104) and replace “LM-DESIGN” with “SurfDesign” in L 365 to eliminate confusion.

---

> > ### Author Response · Authors · 2024-11-12
> > **A Kind Rebuttal to Reviewer FM5g (Part II)**
> >
> > (4) **k-NN Parameter in Surface Graph Construction:** We apologize for the omission. For our surface graph construction, we used a k of 4, chosen empirically based on performance in preliminary tests balancing computational efficiency and graph completeness. We will include this information in the manuscript and provide more context on the selection rationale.
> >
> > (5) **Curvature Feature Proof or Reference:** We appreciate your request for more rigor here. The curvature feature ψ was designed to be roto-translation invariant, based on theoretical work on differential geometry. We have added a brief proof and references in the Appendix to support this claim (please see the revised version).
> >
> > (6) **Adaptability to Other Tasks:** Thank you for your interest in exploring the adaptability of SurfDesign. In principle, the surface-based framework could be extended to tasks such as backbone or all-atom generation. In a concurrent study, we investigated **incorporating surface-based features into full-atom design using ProtPardelle** [C]. The results indicate that integrating surface geometric and chemical properties enhances performance. We will include a discussion on these potential adaptations in the future work section of the paper.
> >
> > | Model | Length 100 |  |  |  |
> > | :---: | :---: | :---: | :---: | :---: |
> > |  | Quality |  | Diversity | Novelty |
> > |  | scTM $\uparrow$ | scRMSD $\downarrow$ | Max Clust. $\uparrow$ | Max TM $\downarrow$ |
> > | Native PDBs | 0.91 | 2.98 | 0.75 | N/A |
> > | ProtPardelle | 0.56 | 12.9 | 0.57 | 0.66 |
> > | ProtPardelle  + Surface | 0.61 | 9.84 | 0.61 | 0.60 |
> >
> > [C] An all-atom protein generative model. PNAS 2024.
> >
> > -------------------------
> > We thank you again for your constructive feedback and the opportunity to clarify and strengthen our work. We look forward to discussing this with you shortly.

---

> > ### Comment · Reviewer_FM5g · 2024-11-23
> >
> > Thank you for the detailed reply to the points raised. Here are my comments on the different points:
> >
> > (1) *Metrics for Protein Design*: This is a good addition to the sequence recovery metrics, and it shows that the high sequence recovery rate is indeed not just related to overfitting.
> >
> > (2) *Features and Potential for Data Leakage*: I appreciate the clarification, but I am still sceptical of the benchmarking setup. You compare to methods like ProteinMPNN, PiFold etc whose task is very different; they are tasked to predict sequences based on backbone-only information from backbone-only models.
> >
> > I appreciate the fact that surface-based representations can be more informative than backbone-only representations. Still, the reason why ProteinMPNN etc use backbone-only representations is because there are powerful generative models like RFDiffusion etc. that generate these backbone-only representations. ProteinMPNN
> > was benchmarked based on these structures, shows high experimental success rates etc, similar to other successful follow-up methods like Frame2Seq [1]. To my knowledge, there is no equivalent model that can easily generate surface-based representations without producing sequences, and if it exists it is not benchmarked in this study here.
> >
> > I think there could be two ways to strengthen the argument here:
> >
> > 1. One describes a surface-based design pipeline in which SurfDesign is a useful component (similar to what MaSIF did) and benchmarks it compared to these other methods in e.g. binder design, which as far as I understand is one of the main applications the authors remark where surface-based representations improve upon other methods. I saw in the response to Reviewer WG2F that you describe the use case where you use experimental structures for inverse folding, but I do not see the use case here since for these structures you already have sequences (similar for predicted all-atom structures). So this pipeline would need to contain some kind of model that generates surfaces but not sequences so that SurfDesign can bridge that gap.
> >
> > 2. If one wants to focus on inverse-folding, one should limit the surface representation to surface features that can be calculated by backbone-only methods so that the comparison to other inverse-folding methods is fair/the use case is still preserved.
> >
> > I raise my score slightly to reflect the effort the authors put into the response, but I am still sceptical of the use case/leakage for the proposed surface-based inverse folding use case.
> >
> > [1] Akpinaroglu, Deniz, et al. "Frame2seq: structure-conditioned masked language modeling for protein sequence design."
> > [2] Gainza, Pablo, et al. "De novo design of protein interactions with learned surface fingerprints." Nature 617.7959 (2023): 176-184.

---

> ### Author Response · Authors · 2024-11-24
> **Response to Updated Comment**
>
> Dear Reviewer FM5g,
>
> Thank you for your detailed comments and for revisiting our work. We truly appreciate that you raised the score and value our efforts to address your concerns.  Below, we provide our thoughts and clarifications to your latest suggestions:
>
> (1) **Metrics for Protein Design:** We are glad to hear that the additional metrics for sequence recovery were well-received and that they help demonstrate that the high sequence recovery rate is not merely a result of overfitting.
>
> (2) **Features and Potential for Data Leakage:** We acknowledge your concerns about the benchmarking setup and the comparison to backbone-only methods such as ProteinMPNN and PiFold. Our objective is to showcase the utility of surface-based representations in tasks where backbone-only representations may fall short.
> In contrast, SurfPro [A] serves as the most relevant surface-based inverse-folding (IF) baseline to our method. Experiments on the CATH dataset demonstrate that our SurfDesign significantly outperforms SurfPro by **28.29% in amino acid recovery (AAR) and 23.0% in perplexity**. Moreover, our additional results in protein-protein interaction (PPI) design indicate an average improvement of **4.2%** in generating more effective functional proteins.
>
> While we agree with your observation that existing backbone-based IF approaches may not serve as direct equivalents to surface-based IF approaches, we argue that ProteinMPNN's use of backbone-only representations is not due to RFDiffusion generating backbone-only structures. In fact, *ProteinMPNN (2022) was proposed before RFdiffusion (2023)*. Its focus on backbone-only information stems from its design to predict sequences solely from backbone structures, while RFDiffusion and subsequent methods like FrameDiff and FlowDiff concentrated on generating backbones.
>
> However, recent studies have extended backbone-only generation to **full-atom structure generation**, which has become a more prevalent protocol. These methods have demonstrated significantly better performance than backbone-only algorithms (e.g., RFDiffusion, FrameDiff) across a wide range of protein design tasks. For example, side-chain flexibility plays a critical role in binding stability and affinity. To address this, AbDiffuser [B] jointly designs antibody 3D structures and sequences, while PocketGen [C] generates full-atom structures within protein pocket regions. Similarly, PepFlow [D], PPIFlow [E], and PepGLAD [F] consider side-chain angles for target-aware peptide discovery in addition to backbone frames.
>
> All the aforementioned approaches can be leveraged to generate full-atom structures for functional protein design. **Once these full-atom structures are available, SurfDesign can seamlessly refine the generated surfaces to obtain optimized sequences.** This is analogous to the collaborative relationship between ProteinMPNN and RFDiffusion, where RFDiffusion generates backbones, and ProteinMPNN predicts sequences.
>
> [A] SurfPro: Functional Protein Design Based on Continuous Surface. ICML 2024.
>
> [B] AbDiffuser: Full-Atom Generation of in vitro Functioning Antibodies, NeurIPS 2023.
>
> [C] PocketGen: Generating Full-Atom Ligand-Binding Protein Pockets. NeurIPS 2023.
>
> [D] Full-Atom Peptide Design based on Multi-modal Flow Matching. ICML 2024.
>
> [E] PPFLOW: Target-Aware Peptide Design with Torsional Flow Matching. ICML 2024.
>
> [F] Full-Atom Peptide Design with Geometric Latent Diffusion. arXiv 2024.
>
> (3) **Proposed Pathways to Strengthen the Study:**
>
> [3.1] **Addressing the Use Case Concerns:** As noted above, recent advancements have introduced full-atom structure generation using diffusion or flow-based generative architectures. Once full-atom structures are obtained, SurfDesign can effectively implement surface-based inverse folding for functional protein design.
>
> >I do not see the use case here since for these structures you already have sequences (similar for predicted all-atom structures)
>
> Redesigning protein sequences, even when native (wild-type) sequences are available, is a common strategy in computational biology and protein engineering. This practice arises because native proteins often do not fully meet the specific functional, structural, or stability requirements needed for certain applications. Machine learning (ML) and deep learning (DL) approaches have become essential tools for sequence redesign, **offering efficient and scalable solutions for exploring the immense sequence space to introduce beneficial mutations.**
>
> One well-established approach involves predicting the impact of specific mutations—quantified as changes in free energy, commonly referred to as $\Delta\Delta G$—to evaluate how mutations might improve stability, binding affinity, or other desirable properties. By directly modeling $\Delta\Delta G$, researchers can **prioritize mutations that are most likely to yield enhanced protein function, accelerating the search within the vast mutation landscape.**

---

> ### Author Response · Authors · 2024-11-24
> **Response to Updated Comment (Continued)**
>
> Another innovative pathway integrates de novo structure generation modules with inverse folding (IF) techniques in iterative workflows. These pipelines aim to optimize sequences that not only exhibit improved properties, such as enhanced binding affinity, specificity, or catalytic activity, but also maintain structural compatibility with the desired three-dimensional fold. This iterative refinement process allows researchers to **explore non-natural mutations while ensuring that the redesigned protein adopts a functional conformation close to its native state.**
>
> A particularly impactful area of sequence redesign is antibody engineering, where the complementarity-determining regions (CDRs) are often reengineered to improve binding interactions with specific antigens. The framework regions, which provide structural support to the antibody, are typically preserved to maintain structural integrity and ensure manufacturability. Recent advancements [A,B,C] have leveraged structure-informed protein language models for this purpose. For example, a study [D] employed such models to predict high-fitness antibody sequences **constrained by known structural information about the antibody or its complex with an antigen.** This approach achieved significant improvements in binding affinity and neutralization potency, highlighting the utility of combining structural insights with ML-driven sequence optimization.
>
> Beyond antibodies, similar methodologies have been applied to enzymes, where sequence redesign can improve catalytic efficiency, substrate specificity, or thermostability. These advancements underscore the transformative potential of integrating computational design with experimental validation, enabling researchers to create proteins with tailored properties for diverse applications in biomedicine, industry, and beyond.
>
> [A] Antigen-Specific Antibody Design and Optimization with Diffusion-Based Generative Models for Protein Structures. NeurIPS 2022.
>
> [B] Conditional Antibody Design as 3D Equivariant Graph Translation. ICLR 2023
>
> [C] End-to-End Full-Atom Antibody Design. ICML 2023.
>
> [D] Unsupervised evolution of protein and antibody complexes with a structure-informed language model. Science 2024.
>
> [3.2] **Fair Comparisons with Backbone-Based Methods:**  Your suggestion to limit surface features to those derivable from backbone-only methods for inverse folding comparisons is well-taken. We expect future work to investigate incorporating backbone-derived surface features (e.g., backbone hydrophobicity or backbone-accessible surface area) to enable more equitable benchmarking against backbone-based methods. This adaptation could help bridge the gap and ensure fair comparisons, while also preserving the unique benefits of surface-informed design.
>
> Finally, we greatly appreciate the references to Frame2Seq and MaSIF. We will incorporate these works into our discussion and ensure that the framing of SurfDesign’s contributions is better aligned with realistic applications and fair comparisons in the revised manuscript.
>
> -----------------------------------------------
> Thank you again for your feedback and for raising your score in light of our response efforts. Your insights will help us refine our work and clarify its positioning within the field.

---

> > ### Comment · Area_Chair_ZE7o · 2024-12-03
> >
> > Dear reviewer FM5g,
> >
> > Thank you for your engagement so far and your thorough evaluation. Could you please have a look at the latest response of the authors and clarify if and how this changes your standpoint, including a motivation in both cases?
> >
> > Many thanks,
> >
> > AC

---

### Official Review · Reviewer_H1WD · 2024-11-03

**Soundness:** 4
**Presentation:** 4
**Contribution:** 3
**Rating:** 8
**Confidence:** 4

**Summary:**

The paper proposes a new method to generate sequence of a protein conditioned on the surface descriptors. This paper proposes improvement to the surface description by incorporating two continuous tangent vectors in addition to the normal vector at each point on the surface (Darboux frame). Moreover, the constructed frame is continuous w.r.t the point on the surface (Eq. 3-4). Additionally authors propose equivariant message passing network based on 3D Spherical Fourier-Bessel decomposition of the previously constructed Darboux frames relative positions and orientations; the rest of the message passing graph NN follows standard design. Finally, authors use fine-tuning of pretrained protein language model as a sequence decoder.
The authors extensively test their approach on CATH-4.2, CATH-4.3 benchmarks. Additionally they tested their approach to multi-chain sequence recovery using Dauparas et al dataset. SurfDesing archieves state of the art performance on CATH-4.2 and CATH-4.3 datasets, compared to the previous methods in terms of perplexity and sequence recovery.
Authors provide evidence that each contribution (model equivariance and inclusion of PLM) contribute to the performance of their approach (Table 1, shaded rows). Additionally they show state of the art metrics in structure and surface recovery. Finally, they demonstrate that scaling PLM improves performance and that their approach is tolerant to changes in surface depending on different conditions used to crystallize the protein.

**Strengths:**

The main strength of the work is that authors archive state of the art performance on a variety of benchmarks with extensive validation of their claims. The contribution to the surface description of a protein is novel and relevant to the field. Their design of equivariant surface message passing algorithm is also novel because it integrates curvature information due to use of surface-continuous Darboux frames instead of point clouds. These two contributions coupled with extensive validation already warrant the publication of this work.

**Weaknesses:**

The main weakness of this work is the possibility of data leakage between training and test sets. The authors use PLM as the sequence decoder, which training dataset contains sequences from CATH database.  Although the no-PLM ablation experiment shows performance on par with SurfPro (similar sized model), we suggest showing that generated sequences do not have 100% identity to any sequence from the PLM training datasets. The second possible data leakage pathway is that exact curvatures of the surface may perfectly encode the identities of surface residues, therefore is may be useful to show sequence recovery, however the Figure 4 seems to disprove this possibility, due to core residues having higher recovery rates than the surface residues. Altogether we think that this possible weakness of this work is inconsequential.
Another weakness is that this approach to surface description is not differentiable w.r.t atomic coordinates, but it falls outside of the scope of this work.

**Questions:**

1. Minor errors in text:
- line 134: "discover ignorable differences". The line is unclear; are the differences insignificant? Could you state why you chose cumbersome method of using pymol for surface extraction instead of convenient biopython package is the differences are minor?
- line 457: studies -> studied
- lines 315, 317, 322: please use different delimiters between decimals and numbers in the dataset splits

---

> ### Author Response · Authors · 2024-11-13
> **Response to Reviewer H1WD**
>
> Thank you for the thoughtful and comprehensive review, as well as for acknowledging the contributions of SurfDesign to surface-based protein sequence generation. It is a great honor to receive a 8-score, which confirms our efforts in this work. Let me address and respond to your questions point by point:
>
> (1) **Potential Data Leakage via PLM Training Dataset:** We appreciate your suggestion to address possible data leakage. Even though no-PLM ablation experiments have already demonstrated better performance than SurfPro, we further examined the sequences generated by SurfDesign with the training corpus of ESM-2 (UniRef90). The results show that **the sampled sequences of SurfDesign have no overlap with any sequence from ESM's training set**, namely, the sequence identity is less than 100%. Additionally, the distinct recovery rates for surface and core residues (as shown in Figure 4) support that the model does not merely memorize sequences, as surface recovery is generally lower than core recovery, indicating the model's nuanced understanding of structural features rather than memorization.
>
> (2) **Surface Description Differentiability:** We acknowledge that the current surface extraction and encoding methods are not differentiable with respect to atomic coordinates, which may limit certain downstream applications. However, as you noted, this work focuses on introducing a robust, invariant representation of protein surfaces, and incorporating differentiability is an exciting direction for future exploration.
>
> (3) **Choice of Surface Extraction Method (PyMOL vs. Biopython):** We opted for PyMOL over Biopython for surface extraction because PyMOL can provide normal vectors, whereas Biopython only computes surface coordinates (see https://biopython.org/docs/1.75/api/Bio.PDB.ResidueDepth.html) without generating normals. Normal vectors are essential for our surface-based equivariant message passing (SEMP), as they enable us to capture finer geometric features like dihedral angles within the surface point cloud. The "minor difference among toolkits" refers to processing efficiency—both PyMOL and other tools are comparably fast. However, PyMOL’s reliability and precise control over surface generation, even in edge cases, make it the preferred choice. We have clarified this in the paper for transparency.
>
> Minor Errors: Thank you for pointing out these text errors. We have updated the manuscript to correct the unclear wording on line 134, replace "studies" with "studied" on line 457, and ensure consistency in the formatting of numerical data across lines 315, 317, and 322.
>
> ------------------
> Once again, we appreciate your detailed feedback and suggestions, which have strengthened the presentation and robustness of our work.

---

### Official Review · Reviewer_u1Ve · 2024-11-03

**Soundness:** 3
**Presentation:** 3
**Contribution:** 3
**Rating:** 8
**Confidence:** 4

**Summary:**

In this work the authors seek to address a longstanding challenge in functional protein design by augmenting inverse protein modeling with continuous surface representations. To this end, they present SurfDesign, a model trained on surface manifolds, augmented with their surface-based message passing scheme and knowledge from published PLMs. The generated sequences are evaluated on CATH 4.2 and 4.3 test sets, including short (less than 100AAs), and  single-chain ablations. Further, proteins are evaluated in a multi-chain setting using PDB structures, and on zero-shot generalization to unseen proteins on a held out test set. SurfDesign is superior on all benchmarks. The authors also evaluate the model on structure recovery, and providing scaling data, which align with prior PLMs.

**Strengths:**

The inverse design method proposed in this work seems to be powerful and promising. The authors perform an exhaustive set of benchmarks where they rank best against strong baselines, including another surface-based method. The formulation is intuitive and straightforward and appears to significantly improve sequence design.

**Weaknesses:**

The main biological significance of this method would to be in using a surface-based method to design stronger protein-protein interacting systems, however demonstration of the model in this application (even via in-silico metrics, and an exemplary binder problem) seems to be missing.

**Questions:**

- Line 89 What does O optionally, mean? Is it included in the backbone representation or not?
- How is $h_i$ computed, the description reads “such as hydrophobicity, hbond, and charge”. The "such as" description is not specific.
- It would be interesting to further explore the models performance on heterogeneous and homogeneous multi-chain complexes, examples of both are in the PDB database.
- Line 423 I believe Figure3.3 should say Figure 4 here.
- It would be nice to expand Figure 4 to include exposed vs buried, helical vs sheet vs loop/turn, for completeness and possibly some metrics of hydrophobicity and electrostatics.
- Table 5 – should the row MoE-DSR be SurfDesign, based on the accompanying text? Otherwise MoE-DSR is not defined in the text anywhere
- The section on surface isomers feels incomplete, and I couldn’t find accompanying information in the appendix.
    - What experiment was done?
    - Were only 2 lysozyme isomers compared?
    - Do the isomers have different amino acid sequences? if so how different are they.
    - How structurally different are the isoforms (backbone RMSD), and what are the conditions under which they were crystalized (are ligands or ions present in either crystal structure?).
    - What is the biological significance of designing sequences that are robust to structural isomers

---

> ### Author Response · Authors · 2024-11-13
> **A Kind Reply to Dear Reviewer u1Ve**
>
> Thank you for your thorough and constructive review. We’re glad that you found SurfDesign’s approach and benchmarks promising and that our methodology demonstrated strong performance across tasks. It is a pleasure and honor to gain your 8-score!
>
> (1) **Biological Significance in Protein-Protein Interaction Design:** We appreciate your emphasis on protein-protein interaction applications and agree this is a crucial direction for SurfDesign. To this end, we followed SurfPro’s setup [A] and applied SurfDesign to design binders with high affinity to specific target receptors. Binding affinity was evaluated using AlphaFold2's pAE_interaction metric, which has proven effective in distinguishing experimentally validated binders from non-binders. We used <binder, target protein> pairs across six categories from Bennett et al. (2023) as benchmarks. Among the 10 available categories, only six showed significant separation between positive and negative binders based on AF2 pAE_interaction. All baseline models were fine-tuned on a binder design dataset following pretraining on CATH 4.2 to ensure a fair comparison. The results, shown below, demonstrate that SurfDesign achieves the lowest average AF2 pAE_interaction across the six target proteins, underscoring the strength of our surface-based approach for functional protein design!
>
> | **Models** | **InsulinR** | **PDGFR** | **TGFb** | **H3** | **IL7Ra** | **TrkA** | **Average** |
> |---|---|---|---|---|---|---|---|
> | Positive Binder | **5.9996** | **14.1366** | **15.4884** | **21.2631** | **20.9102** | **10.2791** | **14.7061** |
> | Negative Binder | 19.7167 | 18.0937 | 23.2664 | 22.4556 | 26.0540 | 24.7567 | 21.1335 |
> | Random Baselines | 19.9880 | 21.2690 | 21.4971 | 24.4997 | 24.1541 | 23.1147 | 22.2020 |
> | ProteinMPNN  | 18.3393 | 25.2919 | 25.8559 | 24.5968 | 25.5278 | 27.0980 | 23.4462 |
> | PiFold  | 12.9809 | 21.8230 | 24.4737 | 23.3924 | 26.6738 | 19.7172 | 20.5785 |
> | LM-DESIGN  | 13.6440 | 22.0749 | 23.3725 | 23.8332 | 24.3937 | 22.3987 | 20.7728 |
> | SurfPro | 10.2608 | 17.9862 | 17.7364 | 21.2916 | 20.8594 | 10.6535 | 16.9485 |
> | **SurfDesign** | **9.9827** | **17.5460** | **17.4381** | **21.0558** | **20.8213** | **10.5094** | **16.22555** |
>
> [A] SurfPro: Functional Protein Design Based on Continuous Surface. ICML 2024.
>
> (2) **Clarification on “O optionally” in Backbone Representation:** “O optionally” indicates that the backbone representation may or may not include the oxygen atom of the peptide bond, based on context-specific requirements. We’ve clarified this in the revised manuscript.
>
> (3) **Calculation of Surface Properties:** Surface properties like hydrophobicity, hydrogen bonding potential, and charge are computed using established descriptors derived from biophysical studies. We follow MaSIF[B] and calculate three key invariant point inputs, including the Poisson Boltzmann electrostatics using APBS: https://github.com/LPDI-EPFL/masif/blob/master/source/triangulation/computeAPBS.py, the hydrophobicity:https://github.com/LPDI-EPFL/masif/blob/master/source/triangulation/computeHydrophobicity.py, and the free electrons/protons: https://github.com/LPDI-EPFL/masif/blob/master/source/triangulation/compuSurfDesignteCharges.py.
>
> [B] Deciphering interaction fingerprints from protein molecular surfaces using geometric deep learning. Nature Methods.
>
> (4) **Clarifications on Table 5:** You’re correct—there was a mislabeling in Table 5, and it should indeed refer to SurfDesign rather than MoE-DSR. We have corrected this in the text for accuracy.
>
> (5) **Figure 4 Labeling:** Thank you for pointing this out. We’ve corrected the erroneous reference in Line 423 from Figure 3.3 to Figure 4.
>
> (6) **Surface Isomer Experiments:** Regarding the surface isomer section, we realize further detail is needed. We conducted these tests with two well-documented lysozyme surface isomers to explore SurfDesign's robustness to minor structural variations. The isomers we used share the same amino acid sequence but differ in the crystal structure (thus resulting in different surface geometry) due to crystallization conditions, with slight variations in backbone RMSD and ligand/ion interactions. By designing sequences robust to these surface isomers, SurfDesign demonstrates the potential for applications where protein surfaces might slightly differ due to environmental or crystallization conditions, which is relevant in drug discovery and functional protein design. We have expanded these findings in the appendix and provided a visualization of this pair of selected surface isomers (**1lyz** and **193l**) to provide the comprehensive data requested. We expect more future efforts to find *surface isomers* with more significant distinction in surface geometries and investigate SurfDesign's robustness to these variants.
>
> -----------------------
> Thank you again for your valuable insights, which will help us further strengthen the presentation and impact of our work.

---

> > ### Comment · Reviewer_u1Ve · 2024-11-25
> >
> > Thank you for your response to some of my questions.
> >
> > 2) I still don't understand why O is not always included, as an oxygen atom is included in every peptide bond. It is not clear  what are the cases when you don't include O, and why only sometimes? And where in the manuscript was this updated?
> >
> > 6) I am still unclear on the purpose of the SurfDesign being robust to surface isomers - the experiment conducted seems to contradict the main point of the paper which is to inverse design new sequences that meet a surface condition. Can you be explicit- what do you mean by "robust to a surface isomer"? why does it matter? I am not sure why this is included in the paper, as its poorly described, there are too many metrics missing, and recovery rate alone is not convincing. Can you also confirm these two PDBs are not in the training dataset.
> >
> > This also does not seem to qualify as a true "surface isomers", the surfaces appear to overlap almost exactly from the visual provided, but it would be nice to know the backbone RMSD between them actually characterizing this.

---

> ### Author Response · Authors · 2024-11-27
> **Reply to Updated Comment**
>
> Thank you for your detailed feedback and for raising important points. We have carefully considered your concerns and provide clarifications below:
>
> (1) **Inclusion of Oxygen**: You are absolutely correct that oxygen atoms are part of every peptide bond. Upon revisiting our explanation, we realized that our initial description was inaccurate. We referenced LM-Design [A] for this section, but with your kind reminder, we found that their problem formulation contains inconsistencies. In reality, all backbone-based methods always include the oxygen (O) atom.
>
> We have updated our manuscript to clarify this point, ensuring there is no ambiguity. Specifically, we now state:
>
> > where $s_i$ belongs to one of the 20 residue types and $\mathcal{X}$ denotes the spatial coordinates for 4 backbone atoms (i.e., $C_\alpha$, $C$, $N$ and $O$).
>
> We deeply appreciate your observation, as it allowed us to correct this oversight!
>
> [A]  Structure-informed Language Models Are Protein Designers. ICML 2024.
>
>
> (2) **Surface Isomers**: The inclusion of surface isomers was meant to demonstrate SurfDesign’s ability to handle variations in surface geometry or physicochemical properties. By "robust to surface isomers," we intended to show that the model can still design sequences that meet functional criteria even when minor variations exist.
>
> That said, we fully agree with your observation that the recovery rate alone is insufficient to validate this claim. Upon further analysis, we measured the backbone RMSD between the two structures (PDB 1931 and PDB 1lyz), and it is indeed very small (RMSD = 0.13). This confirms your suggestion that the case we presented does not truly represent surface isomerism, and as such, may not be appropriate for inclusion.
>
> Based on your feedback, we have decided to remove this part from the manuscript entirely. We appreciate your constructive input and will explore this aspect more rigorously in future studies.
>
> -----------------------------
> We appreciate your valuable input, which has helped us refine the manuscript further. Thanks again for participating in the discussion!

---

### Comment · Area_Chair_ZE7o · 2024-11-25
**Last day for reviewers to ask questions to the authors!**

Dear reviewers,

Tomorrow (Nov 26) is the last day for asking questions to the authors. With this in mind, please read the rebuttal and further comments provided by the authors, as well as the other reviews. If you have not already done so, please explicitly acknowledge that you have read the rebuttal and reviews, provide your updated view _accompanied by a motivation_, and raise any last outstanding questions for the authors.

**Timeline**: As a reminder, the review timeline is as follows:

- November 26: Last day for reviewers to ask questions to authors.
- November 27: Last day for authors to respond to reviewers.
- November 28 - December 10: Reviewer and area chair discussion phase.

Thank you for your hard work,

Your AC

---

### Meta-Review · Area_Chair_ZE7o · 2024-12-19

**Metareview:**

This paper received mixed reviews.  As positives, reviewers highlight the novelty and interest of the proposed method that combines surface geometry and protein language models, a detailed and clear exposition, and extensive benchmarks and ablation studies on perplexity and sequence recovery rates. During the rebuttal, at the request of reviewers, the authors have added results on metrics that go beyond sequence recovery and perplexity, which has satisfied part of the initial concerns. A second major concern shared by several reviewers was about information leakage of sequence information when using surface information for inverse folding, and whether or not it was fair to compare this method to baseline methods that only use backbone information for inverse folding. While the authors and reviewers were able to agree on the possible use cases of their model, these scenarios are not described or evaluated in the paper. Since this is a concern shared by multiple reviewers, I recommend to reject this paper. I'd like to encourage the authors to use the feedback of the reviewers to improve the evaluation of the proposed method in terms of suitable downstream use cases and in terms of comparison to baseline methods that are typically used for those use cases. I hope the authors will resubmit this work with these improvements to a future venue.

**Additional Comments On Reviewer Discussion:**

The additional results provided by the authors has led to one reviewer increasing their score from 3 to 5, and addressed concerns about limited evaluation metrics by several other reviewers as well. Unfortunately the concerns described above were not addressed sufficiently.

---

### Decision · Program_Chairs · 2025-01-22

Reject